# A framework for evaluating clinical artificial intelligence systems without ground-truth annotations

Dani Kiyasseh [1] ✉, Aaron Cohen[2,3], Chengsheng Jiang[2] & Nicholas Altieri[2]

A clinical artificial intelligence (AI) system is often validated on data withheld during its development. This provides an estimate of its performance upon future deployment on data in the wild; those currently unseen but are expected to be encountered in a clinical setting. However, estimating performance on data in the wild is complicated by distribution shift between data in the wild and withheld data and the absence of ground-truth annotations. Here, we introduce SUDO, a framework for evaluating AI systems on data in the wild. Through experiments on AI systems developed for dermatology images, histopathology patches, and clinical notes, we show that SUDO can identify unreliable predictions, inform the selection of models, and allow for the previously out-of-reach assessment of algorithmic bias for data in the wild without ground-truth annotations. These capabilities can contribute to the deployment of trustworthy and ethical AI systems in medicine.

A clinical artificial intelligence (AI) system is often developed to achieve some task (e.g., diagnose prostate cancer[1]) on some training data and subsequently validated on a held-out set of data to which it has never been exposed. This widely-adopted evaluation process assumes that the held-out data are representative of data in the wild[2]; those which are currently unseen yet are expected to be encountered in a clinical setting. For example, an AI system may be trained on data from one electronic health record (EHR) system and subsequently deployed on data from another EHR system. However, data in the wild often (a) follow a distribution which is different from that of the held-out data and (b) lack ground-truth labels for the task at hand (Fig. 1a). Combined, such distribution shift which is known to adversely affect the behaviour of AI systems[3], and the absence of ground-truth labels complicate the evaluation of an AI system and its predictions. It becomes challenging to identify reliable AI predictions, select favourable AI systems for achieving some task, and even perform additional checks such as assessing algorithmic bias[4]. Incorrect predictions, stemming from data distribution shift, can lead to inaccurate decisions, decreased trust, and potential issues of bias. As such, there is a pressing need for a framework that enables more reliable decisions in the face of AI predictions on data in the wild.

To address this need, previous work assumes highly-confident predictions are reliable[5,6], even though AI systems are known to generate highly-confident incorrect predictions[7]. Recognising these limitations, others have demonstrated the value of modifying AI-based confidence scores through explicit calibration methods such as Platt scaling[8,9] or through ensemble models[10]. Such calibration methods, however, can be ineffective when deployed on data in the wild that exhibit distribution shift[11]. Regardless, quantifying the effectiveness of calibration methods would still require ground-truth labels, a missing element of data in the wild. Another line of research focuses on estimating the overall performance of models with unlabelled data[12,13]. However, it tends to be model-centric, overlooking the data-centric decisions (e.g., identifying unreliable predictions) that would need to be made upon deployment of these models, and makes the oft fallible assumption that the held-out set of data is representative of data in the wild, and therefore erroneously extends findings in the former setting to those in the latter.

In this study, we propose pseudo-label discrepancy (SUDO), a framework for evaluating AI systems deployed on data in the wild. Through experiments on three clinical datasets (dermatology images, histopathology patches, and clinical notes), we show that SUDO can be

[1]Cedars-Sinai Medical Center, Los Angeles, CA, USA. [2]Flatiron Health, New York City, NY, USA. [3]New York University School of Medicine, New York City, NY, USA. ✉e-mail: danikiy@hotmail.com

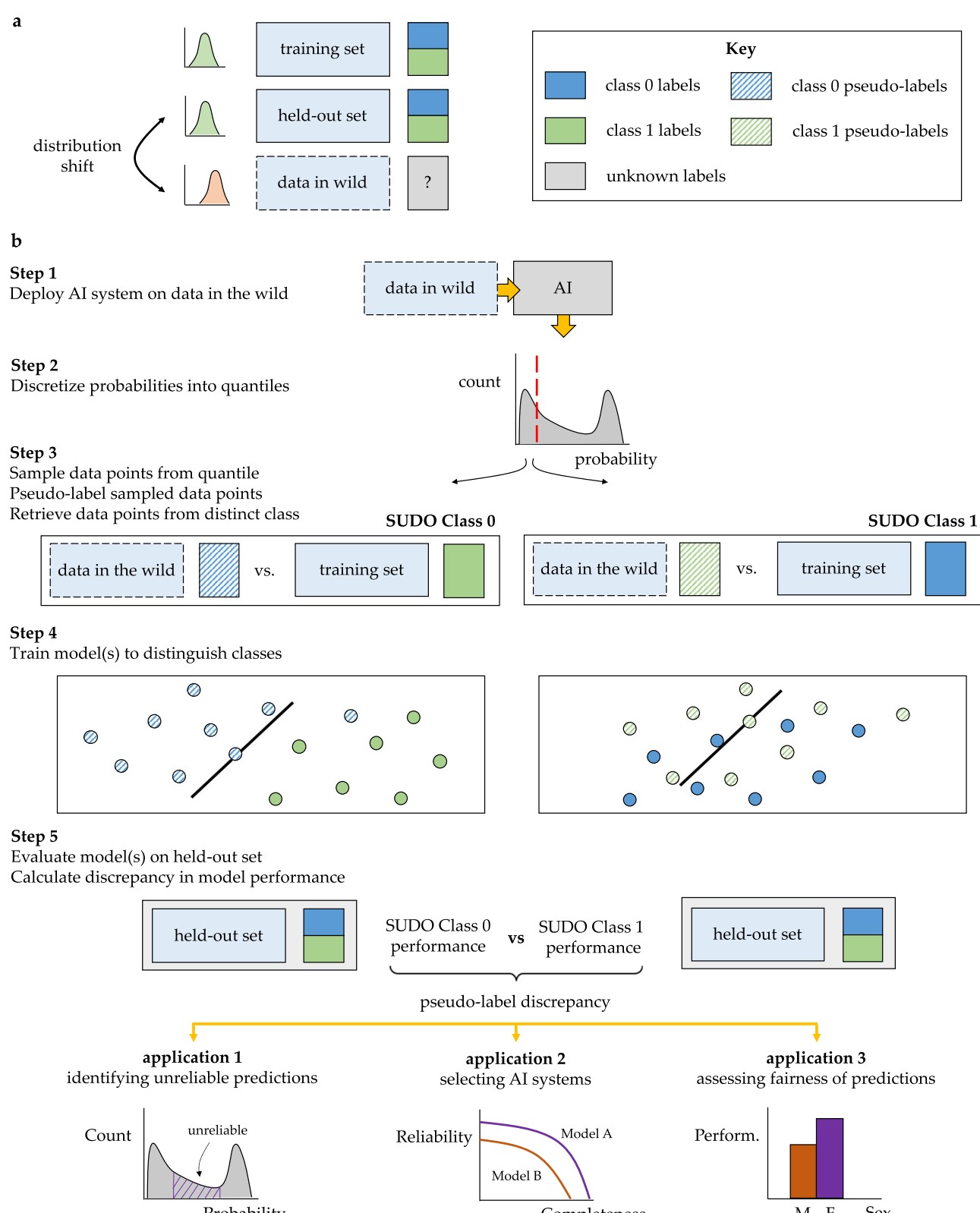

**Fig. 1 | SUDO is a framework to evaluate AI systems without ground-truth labels. a** An AI system is often deployed on data in the wild, which can vary significantly from those in the held-out set (distribution shift), and which can also lack ground-truth labels. **b** SUDO is a 5-step framework that circumvents the challenges posed by data in the wild. First, deploy an AI system on data in the wild to obtain probability values. Second, discretize those values into quantiles. Third, sample data points from each quantile and pseudo-label (temporarily label) them with a possible class (SUDO Class 0). Sample data points with ground-truth labels from the opposite class to form a classification task. Fourth, train a classifier to distinguish between these data points. Repeat the process with a different pseudo-label (SUDO Class 1). Finally, evaluate and compare the performance of the classifiers on the same held-out set of data with ground-truth labels, deriving the pseudo-label discrepancy. This discrepancy and the relative classifier performance indicate whether the sampled data points are more likely to belong to one class than another.

a reliable proxy for model performance and thus be used to identify unreliable AI predictions. This finding holds even with overconfident models. We also show that SUDO informs the selection of models upon their deployment on data in the wild. By implementing SUDO across patient groups, we demonstrate that it also allows for the previously out-of-reach assessment of algorithmic bias for data without ground-truth labels.

## Results

### Overview of the SUDO framework
SUDO is a framework that helps identify unreliable AI predictions, select favourable AI systems, and assess algorithmic bias for data in the wild without ground-truth labels. We outline the mechanics of SUDO through a series of steps (Fig. 1b).

**Step 1 ·** Deploy probabilistic AI system on data points in the wild and return probability, $s \in [0, 1]$, of positive class for each data point.

**Step 2 ·** Generate distribution of output probabilities and discretize them into several predefined intervals (e.g., deciles).

**Step 3 ·** Sample data points in the wild from each interval and assign them a temporary class label (pseudo label). Retrieve an equal number of data points with the opposite class label from the training set (ground-truth).

**Step 4 ·** Train a classifier to distinguish between the pseudo-labelled data points and those with a ground-truth label.

**Step 5 ·** Evaluate classifier on held-out set of data with ground-truth labels (e.g., using any metric such as AUC). A performant classifier supports the validity of the pseudo-label. However, each interval may consist of data points from multiple classes, exhibiting class contamination. To detect this contamination, we repeat these steps while cycling through the different possible pseudo-labels.

**Pseudo-label discrepancy ·** Calculate the discrepancy between the performance of the classifiers with different pseudo labels. The greater the discrepancy between classifiers, the lower the class contamination, and the more likely that the data points belong to a single class. We refer to this discrepancy as the pseudo-label discrepancy or SUDO.

### SUDO correlates with model performance on Stanford diverse dermatology images dataset
We used SUDO to evaluate predictions made on the Stanford diverse dermatology image (DDI) data[14] (*n*: 656) (see Description of datasets). We purposefully chose two AI models (DeepDerm[15] and HAM10000[16]) that were performant on their respective data (AUC = 0.88 and 0.92) and whose performance degraded drastically when deployed on the DDI data (AUC = 0.56 and 0.67), suggesting the presence of distribution shift.

We found that these models struggle to distinguish between benign (negative) and malignant (positive) lesions in images. This is evident by the lack of separability of the AI-based probabilities corresponding to the ground-truth negative and positive classes (Fig. 2a for DeepDerm and Fig. 2b for HAM10000). We set out to determine whether SUDO, without having access to the ground-truth labels, can quantify this class contamination. We found that SUDO correlates with the proportion of positive instances in each of the chosen probability intervals ($\rho = -0.84 \, p < 0.005$ for DeepDerm in Fig. 2c, and $\rho = -0.76 \, p < 0.01$ for HAM10000 in Fig. 2d). Such a finding, which holds regardless of the evaluation metric used (Supplementary Fig. 5), suggests that SUDO can be a reliable proxy for the accuracy of predictions. Notably, this ability holds irrespective of the underlying performance of the AI model being evaluated, as evidenced by the high correlation values for the two models which performed at different levels (AUC = 0.56 and 0.67).

### SUDO informs model selection on Stanford diverse dermatology images dataset
We can leverage SUDO to create two tiers of predictions. Reliable predictions are associated with large SUDO values that indicate low class contamination and are therefore incorporated into downstream analyses. The remaining predictions are considered unreliable and flagged for further review by a human expert. By changing our threshold for reliable predictions, we notice a trade-off between the reliability of such predictions and the proportion of which is incorporated into downstream analyses (i.e., completeness). For example, by selecting only the most reliable predictions, we reduce their completeness. Ideally, models should produce predictions that exhibit both high reliability and completeness. These two dimensions, which we capture via the reliability-completeness curve (see Producing reliability-completeness curve in Methods for details), offer an opportunity to rank order models particularly when ground-truth labels are unavailable (Fig. 2e).

We found that the ordering of the performance of the models is consistent with that presented in previous studies[14]. Specifically, HAM10000 and DeepDerm achieve an area under the reliability-completeness curve of AURCC = 0.86 and 0.62, respectively and, with ground-truth annotations, these models achieve (AUC = 0.67 and 0.56). We note that the emphasis here is on the relative ordering of models and not on their absolute performance. These consistent findings suggest that SUDO can help inform model selection on data in the wild without annotations.

### SUDO helps assess algorithmic bias without ground-truth annotations
Algorithmic bias often manifests as a discrepancy in model performance across two protected groups (e.g., male and female patients). Traditionally, this would involve comparing AI predictions to ground-truth labels. With SUDO as a proxy for model performance, we hypothesised that it can help assess such bias even without ground-truth labels. We tested this hypothesis on the Stanford DDI dataset by stratifying the AI predictions according to the skin tone of the patients (Fitzpatrick scale I-II vs. V-VI) and implementing SUDO for each of these stratified groups. A difference in the resultant SUDO values would indicate a higher degree of class contamination (and therefore poorer performance) for one group over another. We found that $SUDO_{AUC} = 0.60$ and 0.58 for the two groups, respectively. This discrepancy, calculated without ground-truth labels, is indicative of the bias we also observed when using ground-truth labels and the negative predictive value of the predictions (NPV) (NPV = 0.83 and 0.78, respectively). Our findings demonstrate that both SUDO and the traditional approach (with ground-truth labels) identified a bias in favour of patients with a Fitzpatrick scale of I-II, which is consistent with previously-reported bias findings[14].

### SUDO correlates with model performance on Camelyon17-WILDS histopathology dataset
We provide further evidence that SUDO can identify unreliable predictions on datasets that exhibit distribution shift. Here, we trained a model on the Camelyon17-WILDS dataset to perform binary tumour classification (presence vs. absence) based on a single histopathological image, and evaluated the predictions on the corresponding test set (*n*: 85,054) (see Description of datasets). This dataset has been constructed such that the test set contains data from a hospital unseen during training, and is thus considered in the wild. We found that the trained model achieved an average accuracy ≈ 0.85 despite being presented with images from an unseen hospital (Fig. 3a). We used SUDO to quantify the class contamination across probability intervals (Fig. 3b), and found that it continues to correlate ($\rho = -0.79 \, p < 0.005$) with the proportion of positive instances in each of the intervals.

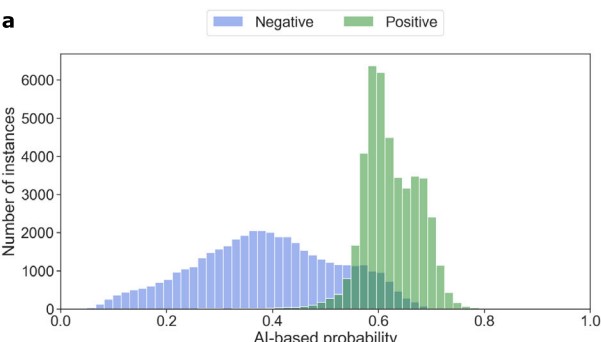
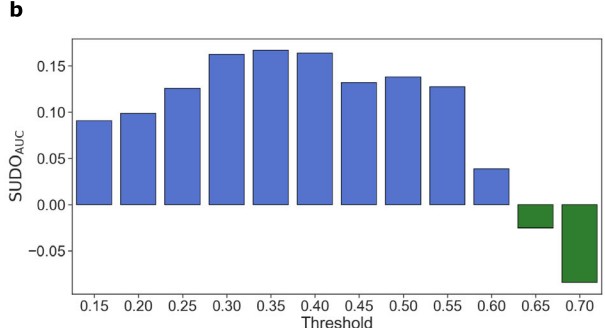

**Fig. 2 | SUDO can be a reliable proxy for model performance on the Stanford diverse dermatology image dataset.** Two models (left column: DeepDerm, right column: HAM10000) are pre-trained on the HAM10000 dataset and deployed on the entire Stanford DDI dataset. **a**, **b** Distribution of the prediction probability values produced by the two models colour-coded based on the ground-truth label (negative vs. positive) of the data points. **c**, **d** Correlation of SUDO with the proportion of positive data points in each probability interval: $|\rho| = 0.94$ ($p < 0.005$) and $|\rho| = 0.76$ ($p < 0.008$), respectively. P-values are calculated based on a two-sided t-test. Results are shown for ten mutually-exclusive probability intervals that span the range [0, 1]. A strong correlation indicates that SUDO can be used to identify unreliable predictions. **e** Reliability-completeness curves of the two models, where the area under the reliability-completeness curve (AURCC) can inform the selection of an AI system without ground-truth annotations. Source data are provided as a Source Data file.

**Fig. 3 | SUDO can be a reliable proxy for model performance on the Camelyon17-WILDS histopathology dataset. a** Distribution of the prediction probability values produced by a model colour-coded based on the ground-truth label (negative vs. positive) of the data points. **b** SUDO values colour-coded according to the most likely label of the predictions in each probability interval. Source data are provided as a Source Data file.

## SUDO can even be used with overconfident models

AI systems are prone to producing erroneous overconfident predictions, complicating our dependence on their confidence scores alone to identify unreliable predictions. It is in these settings where SUDO adds most value. To demonstrate this, we first trained a natural language processing (NLP) model to distinguish between negative ($n$: 1000) and positive ($n$: 1000) sentiment in product reviews with distribution shift as part of the Multi-Domain Sentiment dataset[17] (see Description of datasets). We showed that SUDO continues to correlate with model performance, pointing to the applicability of the framework across data modalities. To simulate an overconfident model, we then overtrained (i.e., extended training for an additional number of epochs) the same NLP model, as confirmed by the more extreme distribution of the prediction probability values (Supplementary Fig. 1b). Notably, we found that SUDO continues to correlate well with model performance despite the presence of overconfident predictions (Supplementary Fig. 1h). This is because SUDO leverages pseudo-labels to quantify class contamination and is not exclusively dependent on confidence scores.

## Exploring the limits of SUDO with simulated data

To shed light on the scenarios in which SUDO remains useful, we conducted experiments on simulated data that we can finely control (see Description of datasets). Specifically, we varied the data in the wild to encompass distribution shift (a) with the same two classes observed during training, (b) with a severe imbalance (8:1) in the number of data points from each class, and (c) alongside data points from a third and never-seen-before class. As SUDO is dependent on the evaluation of classifiers on held-out data (see Fig. 1, Step 5), we also experimented with injecting label noise into such data.

We found that SUDO continues to strongly correlate with model performance, even in the presence of a third class ($|\rho| > 0.87\ p < 0.005$, Supplementary Fig. 2). This is not surprising as SUDO is designed to simply quantify class contamination in each probability interval, regardless of the data points contributing to that contamination. However, we did find that SUDO requires held-out data to exhibit minimal label noise, where $\rho = 0.99 \rightarrow 0.33$ upon randomly flipping 50% of the labels in the held-out data to the opposite class. We also found that drastically changing the relationship between class-specific distributions of data points in the wild can disrupt the utility of SUDO (Supplementary Fig. 3).

## SUDO correlates with model performance on Flatiron Health ECOG Performance Status dataset

To demonstrate the applicability of SUDO to a range of datasets, we investigated whether it also acts as a reliable proxy for model performance on the Flatiron Health Eastern Cooperative Oncology Group Performance Status (ECOG PS) dataset. Building on previous work[18], we developed an NLP model to infer the ECOG PS, a value reflecting a patient's health status, from clinical notes of oncology patient visits (see Methods for description of data and model).

In this section, we exclusively deal with data which (a) do not exhibit distribution shift and (b) are associated with a ground-truth label. The motivation behind these experiments was to first demonstrate that we can learn an NLP model that accurately classifies ECOG PS as a prerequisite for applying SUDO to the target setting in which distribution shift exists and ground-truth labels do not.

We found that the NLP model performs well in classifying ECOG PS from clinical notes of oncology patient visits (precision = 0.97, recall = 0.92, and AUC = 0.93). We hypothesise that these results are driven by the discriminative pairs of words that appear in clinical notes associated with low and high ECOG PS. For example, typical phrases found in low ECOG PS clinical notes include "normal activity" and "feeling good" whereas those found in high ECOG PS clinical notes include "bedridden" and "cannot carry". This strong discriminative

behaviour can be seen by the high separability of the two prediction probability distributions (Fig. 4a). Although it was possible to colour-code these distributions and glean insight into the degree of class contamination, this is not possible in the absence of ground-truth labels. SUDO attempts to provide this insight despite the absence of ground-truth labels.

We found that data points with $s \approx 0$ are more likely to belong to the low ECOG PS label than to the high ECOG PS label, and vice versa for data points with $s \approx 1$. This is evident by the large absolute $SUDO_{AUC}$ values at either end of the probability spectrum (Fig. 4c). This is not surprising and is in line with expectations. Consistent with findings presented earlier, SUDO also correlates with model performance on this dataset. This can be seen by the strong correlation ($|\rho| = 0.95\ p < 0.005$) between SUDO and the proportion of positive instances in each of the chosen probability intervals. This bodes well for when we ultimately use SUDO to identify unreliable predictions without ground-truth annotations.

## Sensitivity analysis of SUDO's hyperparameters

To encourage the adoption of SUDO, we conducted several experiments on the Flatiron Health ECOG PS dataset to measure SUDO's sensitivity to hyperparameters. Specifically, we varied the number of data points sampled from each probability interval (Fig. 1b, Step 3), the type of classifier used to distinguish between pseudo- and ground-truth labelled data points (Fig. 1b, Step 4), and the amount of label noise in the held-out data being evaluated on (Fig. 1b, Step 5). We found that reducing the number of sampled data points (from 200 to just 50) and using different classifiers (logistic regression and random forest) continued to produce a strong correlation between SUDO and model performance ($|\rho| > 0.94\ p < 0.005$) (Supplementary Fig. 4). Such variations, however, altered the directionality (net positive or negative) of the SUDO values (from one experiment to the next) in the probability intervals with a high degree of class contamination. For example, in the interval $0.20 < s < 0.25$ (Fig. 4a), $SUDO_{AUC} = 0.05$ and $SUDO_{AUC} = -0.05$ when sampling 50 and 200 data points, respectively. We argue that such an outcome does not practically affect the interpretation of SUDO, as it is the absolute value of SUDO that matters most when it comes to identifying unreliable predictions. We offer guidelines on how to deal with this scenario in a later section.

## Using SUDO to identify unreliable predictions on Flatiron Health ECOG Performance Status dataset

To further illustrate the utility of SUDO, we deployed the NLP model on the Flatiron Health ECOG PS data in the wild without ground-truth annotations. It is likely that such data (clinical notes without ECOG PS labels) follow a distribution that is distinct from that of the training data (clinical notes with ground-truth ECOG PS labels). This is supported by the distinct distributions of the prediction probability values across these datasets (see Fig. 4a, b). Such a shift can make it ambiguous to identify unreliable predictions based exclusively on confidence scores. To resolve this ambiguity, we implemented SUDO for ten distinct probability intervals, choosing more granular intervals in the range $0 < s < 0.40$ to account for the higher number of predictions (Fig. 4d). These results suggest that predictions with $0 < s < 0.20$ are more likely to belong to the low ECOG PS class than to the high ECOG PS class. The opposite holds for predictions with $0.30 < s < 1$. Such insight, which otherwise would have been impossible without ground-truth annotations, can now better inform the identification of unreliable predictions.

## Validating SUDO-guided predictions with a survival analysis

To gain further confidence in SUDO's ability to identify unreliable predictions, we leveraged the known relationship between ECOG PS and mortality: patients with a higher ECOG PS are at higher risk of mortality[19]. As such, we can compare the overall survival estimates of

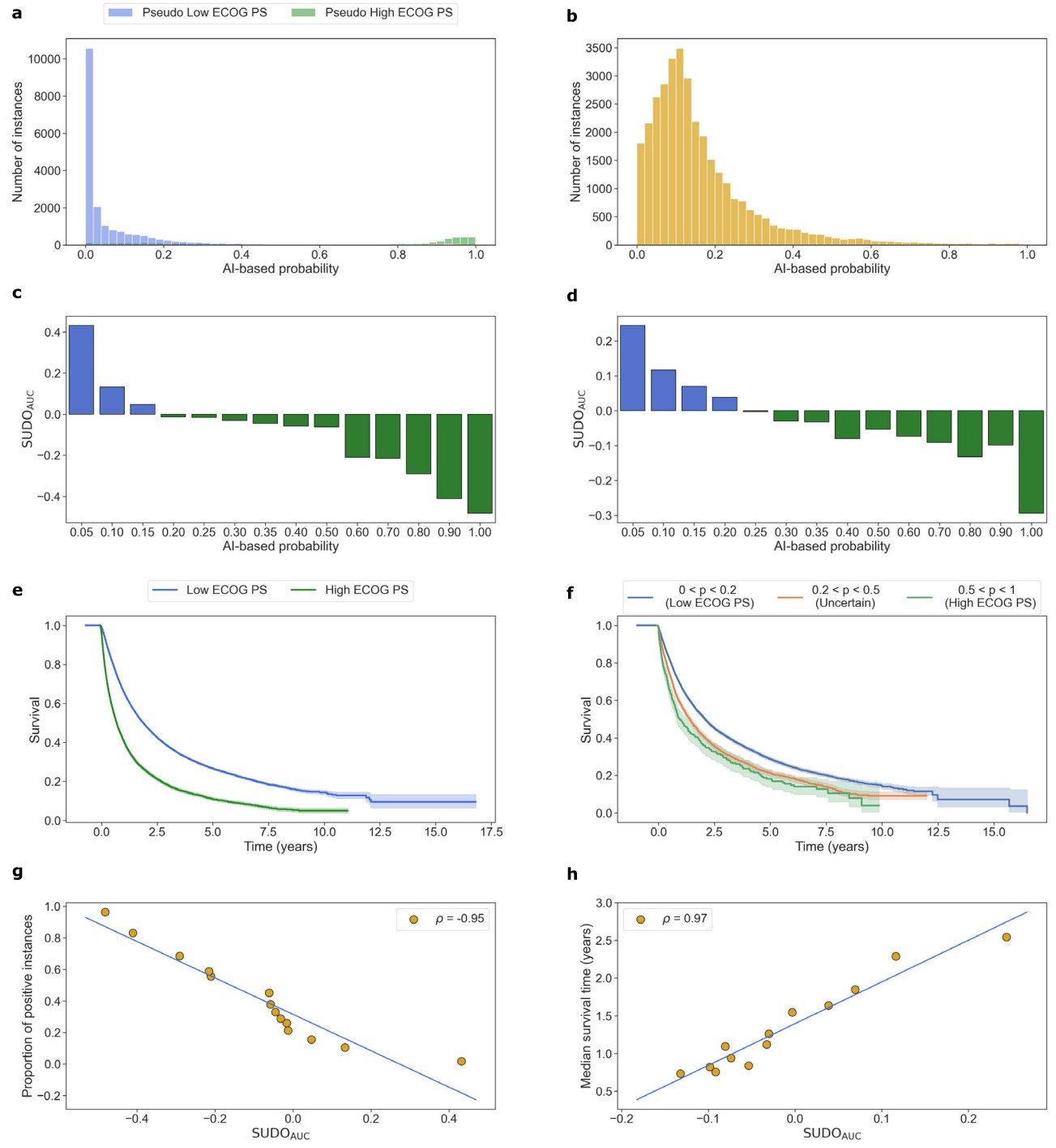

**Fig. 4 | SUDO correlates with model performance on the Flatiron Health ECOG Performance Status data without ground-truth annotations.** Results for (left column) test set with ground-truth annotations and (right column) data in the wild without ground-truth annotations. **a, b** Distribution of prediction probability values of NLP model. **c, d** SUDO values colour-coded with most likely label in each probability interval. Survival curves for patient groups identified via (**e**) ground-truth annotations and (**f**) SUDO values with low ECOG PS (0 < p < 0.2), high ECOG PS

(0.5 < p < 1.0), and unreliable (0.2 < p < 0.5) predictions. The shaded area reflects the 95% confidence interval. **g, h** Correlation between SUDO and proportion of positive instances (using ground-truth annotations, $|\rho| = 0.95\, p < 0.005$) and the median survival time of patients (without ground-truth annotations, $|\rho| = 0.97\, p < 0.005$) in each probability interval. P-values are calculated based on a two-sided t-test. ECOG Eastern Cooperative Oncology Group, PS Performance Status.

patients with reliable AI predictions (i.e., large SUDO values) to those of patients with known ECOG PS values (e.g., patients in the training set). The intuition is that if such overall survival estimates are similar to one another, then we can become more confident in the ECOG PS labels that were newly assigned to clinical notes from oncology patient visits. While this approach makes the assumption that the ECOG PS label is the primary determinant of overall survival, we acknowledge

that additional confounding factors, beyond the ECOG PS label, may also play a role[20].

For patients in the training set of the Flatiron Health ECOG PS dataset, we present their survival curves stratified according to whether they have a low or high ECOG PS (Fig. 4e). For patients in the data in the wild, for whom we do not have a ground-truth ECOG PS label, we first split them into three distinct groups based on the SUDO value (Fig. 4d),

employing the intuition that a higher absolute value is reflective of more reliable predictions (e.g., $|SUDO_{AUC}| > 0.05$). We refer to these groups based on their corresponding predictions: low ECOG PS group ($0 < s \leq 0.2$, $n$: 12, 603), high ECOG PS group ($0.5 \leq s \leq 1.0$, $n$: 552), and an uncertain ECOG PS group ($0.2 < s < 0.5$, $n$: 3729). As demonstrated in an earlier section, the chosen SUDO threshold creates a trade-off between the reliability and completeness of the predictions. We present the group-specific survival curves (Fig. 4f). To control for confounding factors, we only considered data samples associated with the first line of therapy where patients are provided their first medication in their treatment pathway (see Methods for more details).

There are two main takeaways. First, and in alignment with our expectations and established clinical observations, we found that patients in the low ECOG PS group do indeed exhibit a longer median survival time than patients in the high ECOG PS group (1.87 vs. 0.68 years, respectively) (Fig. 4f). Second, the chosen probability intervals based on which the survival analysis was stratified reasonably identified distinct patient cohorts. This is evident by the distinct survival curves of the patient cohorts with $0 < s \leq 0.2$ and $0.5 \leq s < 1$ and their similarity to the survival curves of patients with a ground-truth ECOG PS label (Fig. 4e). For example, the median survival estimates of these two patient cohorts are 2.07 (vs. 1.87) and 0.95 (vs. 0.68) years, respectively. Although we do not expect such values to be perfectly similar, due to potential hidden confounding factors we cannot control for, they are similar enough to suggest that these newly-identified patient cohorts correspond to low and high ECOG PS patient cohorts.

Demonstrating that SUDO correlates with a meaningful variable can engender trust in its design. When ground-truth annotations are available, we chose this variable to be the proportion of positive instances in each probability interval (i.e., accuracy of predictions). Without ground-truth annotations, we chose the median survival time of patients in each interval. Specifically, we quantified the correlation between SUDO and the median survival time of patient cohorts in each of the ten chosen probability intervals (Fig. 4h). We found that that these two variables are indeed strongly correlated ($|\rho| = 0.97$ $p < 0.005$). Such a finding suggests that SUDO can provide useful insight into the clinical characteristics of patient cohorts in datasets without ground-truth labels (see Discussion for benefits and drawbacks of such an approach).

## Practical guidelines for using SUDO

We have made the case and presented evidence that SUDO can evaluate AI systems without ground-truth annotations. We now take stock of our findings to offer practical guidelines around SUDO. First, we demonstrated that SUDO works well across multiple data modalities (images, text, simulation). We therefore recommend using SUDO irrespective of the modality of data a model is evaluated on. Second, we showed that SUDO is agnostic to the neural network architecture of the AI system being evaluated (convolutional for images, feed-forward for text). The only requirement is that the neural network returns a probabilistic value. Third, we showed that SUDO can deal with as few as 50 data points sampled from each probability interval (on the Stanford DDI dataset). Although sampling too few data points did not change the absolute value of SUDO, and thereby reliably quantifying class contamination, it did alter its directionality (negative or positive), affecting the perceived proportion of the majority class in a set of predictions. To avoid being misled by this behaviour, we recommend sampling at least 50% of the data points in each probability interval in order to capture a representative set of predictions. We also note that the absolute value of SUDO should take precedent for determining unreliable predictions. Only if that value is large enough (i.e., low class contamination) should its directionality be considered.

Fourth, we showed that SUDO is unperturbed by an imbalance in the number of data points from each class or by the presence of a third-and-unseen class (on the simulated dataset). If data in the wild are suspected to exhibit these features, then SUDO can still be used. Fifth, we showed that SUDO is sensitive to the quality of the labels in the held-out set of data. As such, we recommend curating a dataset with minimal label noise when using SUDO. Furthermore, we showed that SUDO produces consistent results irrespective of the classifier used to distinguish between pseudo-labelled and ground-truth data points and of the metric used to evaluate these classifiers. We therefore recommend using a lightweight classifier (to speed up computation) and the metric most suitable for the task at hand.

## Discussion

AI systems have long been validated on withheld data, with the assumption that such data are representative of data in the wild. When this assumption is violated, as is often the case with clinical data, and ground-truth annotations are unavailable, it becomes difficult to trust the predictions made by an AI system.

We have shown that SUDO can comfortably assess the reliability of predictions of AI systems deployed on data in the wild. Notably, we demonstrated that SUDO can supplement confidence scores to identify unreliable predictions, help in the selection of AI systems, and assess the algorithmic bias of such systems despite the absence of ground-truth annotations. Although we have presented SUDO primarily for clinical AI systems and datasets, we believe its principles can be applied to probabilistic models in almost any other scientific discipline.

Compared with previous studies, our study offers a wider range of applications for predictions on data without ground-truth annotations. These applications include identifying unreliable predictions, selecting favourable models, and assessing algorithmic bias. Previous work tends to be more model-centric than SUDO, focusing on estimating model performance[12,13,21,22] and assessing algorithmic bias[23] using both labelled and unlabelled data. It therefore overlooks the myriad data-centric decisions that would need to be made upon deployment of an AI system, such as identifying unreliable predictions. The same limitation holds for other studies that attempt to account for verification bias[24,25], a form of distribution shift brought about by only focusing on labelled data. In contrast, SUDO provides the optionality of guiding decisions at the model level (e.g., relative model performance) and at the data level (e.g., identifying unreliable predictions).

Most similar to our work is the concept of reverse testing[26] and reverse validation[27,28] where the performance of a pair of trained AI systems is assessed by deploying them on data without annotations, pseudo-labelling these data points, and training a separate classifier to distinguish between these data points. The classifier that performs better on a held-out set of labelled data is indicative of higher quality pseudo-labels and, by extension, a favourable AI system. SUDO differs from this line of work in two main ways. First, reverse testing assigns a single AI-based pseudo-label to each data point in the wild whereas we assign all possible pseudo-labels to that data point (through distinct experiments) in order to determine the most likely ground-truth label. Second, given the probabilistic output of an AI system, reverse testing performs pseudo-labelling for data points that span the entire probability spectrum for the exclusive purpose of model selection. As such, it cannot be used for identifying unreliable predictions. Notably, previous work heavily depends on the assumption that the held-out set of data is representative of data in the wild. SUDO circumvents this assumption by operating directly on data in the wild.

SUDO's ability to identify unreliable predictions has far-reaching implications. From a clinical standpoint, data points whose predictions are flagged as unreliable can be sent for manual review by a human expert. By extension, and from a scientific standpoint, this layer of human inspection can improve the integrity of research findings. We note that SUDO can be extended to the multi-class setting (e.g., $c > 2$ classes) by cycling through all of the pseudo-labels and retrieving data points from the mutually-exclusive classes (Fig. 1b, Step 3) to train a

total of *c* classifiers (Fig. 1b, Step 4). The main difference to the binary setting is that SUDO would now be calculated as the maximum difference in performance across all classifiers (Fig. 1b, Step 5). SUDO's ability to select favourable (i.e., more performant) AI systems can lead to the deployment of more accurate systems that contribute to improved patient care. SUDO's ability to assess algorithmic bias, which was not previously possible for data without ground-truth labels, can contribute to the ethical deployment of AI systems. This ensures that AI systems perform as expected when deployed on data in the wild. We note that SUDO can also be used to assess algorithmic bias across multiple groups by simply implementing SUDO for data points from each group. Bias would still manifest as a discrepancy in the SUDO value across the groups. Overall, our study offers a first step towards a framework of inferring clinical variables which suffer from low completeness in the EHR (such as ECOG PS) in the absence of explicit documentation in their charts and ground-truth labels.

There are several challenges that our work has yet to address. First, SUDO cannot identify the reliability of a prediction for an individual data points. This is because we often calculated SUDO as a function of prediction probability intervals. This is in contrast to previous work on uncertainty quantification and selective classification. While SUDO can be applied to individual data points, this is not practical as it depends on the learning of predictive classifiers, which typically necessitate a reasonable number of training data points. We do note, though, that SUDO was purposefully designed to assess the relative reliability of of predictions across probability intervals. It is also worth noting that SUDO may be considered excessive if the amount of data in the wild is small and can be annotated by a team of experts with reasonable effort. However, when presented with large-scale data in the wild, SUDO can yield value by acting as a data triage mechanism, funneling the most unreliable predictions for further inspection by human annotators. In doing so, it stands to reduce the annotation burden placed on such annotators. Furthermore, despite having presented evidence of SUDO's utility on multiple real-world datasets with distribution shift, we have not explored how SUDO would behave for the entire space of possible distribution shifts. It therefore remains an open question whether a particular type of distribution shift will render SUDO less meaningful. On some simulated data, for example, we found that SUDO is less meaningful upon introducing drastic label noise or changing the class-specific distributions of the data points in the wild. More generally, we view SUDO merely as one of the first steps in informing decision-making processes. Subsequent analyses, such as statistical significance tests, would be needed to gain further confidence in the resulting conclusions.

To validate SUDO without ground-truth annotations, we measured its correlation with median survival time, a clinical outcome with a known relationship to ECOG PS. This approach was made possible by leveraging domain knowledge. In settings where such a relationship is unknown, we recommend identifying clinical features in the labelled data that are unique to patient cohorts. These features can include the type and dosage of medication patients receive and whether or not they were enroled in a clinical trial. A continuous feature (e.g., medication dosage) may be preferable to a discrete one (e.g., on or off medication) in order to observe a graded response with the prediction probability intervals. If identifying one such feature is difficult and time-consuming, a data-driven alternative could involve clustering patients in the labelled data according to their clinical characteristics. Distinct clusters may encompass a set of features unique to patient cohorts. Prediction on data in the wild can then be assessed based on the degree to which they share these features. On the other hand, the more severe the distribution shift, the less likely it is that features will be shared across the labelled and unlabelled data.

There are also important practical and ethical considerations when it comes to using SUDO. Without SUDO, human experts would have to painstakingly annotate all of the data points in the wild. Such an approach does not scale as datasets grow in size. Moreover, the ambiguity of certain data points can preclude their annotation by human experts. SUDO offers a way to scale the annotation process while simultaneously flagging unreliable predictions for further human inspection. However, as with any AI-based framework, over-reliance on SUDO's findings can pose risks particularly related to mislabelling data points. This can be mitigated, in some respects, by choosing a more conservative operating point on the reliability-completeness curve.

Moving forward, we aim to expand the application areas of SUDO to account for the myriad decisions that AI predictions inform. This could include using SUDO to detect distribution shift in datasets, thereby informing whether, for example, an AI system needs to be retrained on updated data. Another line of research includes improving the robustness of SUDO to label noise and expanding its applicability to scientific domains in which label noise is rampant. We look forward to seeing how the community leverages SUDO for their own applications.

## Methods
### Description of datasets
**Stanford diverse dermatology images.** The Stanford diverse dermatology images (DDI) dataset consists of dermatology images collected in the Stanford Clinics between 2010 and 2020. These images ($n$: 656) reflect either a benign or malignant skin lesion from patients with three distinct skin tones (Fitzpatrick I-II, III-IV, V-VI). For further details, we refer interested readers to the original publication[14]. We chose this as the data in the wild due to a recent study reporting the degradation of several models' performance when deployed on the DDI dataset. These models (see Description of models) were trained on the HAM10000 dataset, which we treated as the source dataset.

**HAM10000 dataset.** The HAM10000 dataset consists of dermatology images collected over 20 years from the Medical University of Vienna and the practice of Cliff Rosendahl[16]. These images ($n$: 10015) reflect a wide range of skin conditions ranging from Bowen's disease and basal cell carcinoma to melanoma. In line with a recent study[14], and to remain consistent with the labels of the Stanford DDI dataset, we map these skin conditions to a binary benign or malignant condition. We randomly split this model into a training and held-out set using a 80: 20 ratio. We did not use a validation set as publicly-available models were already available and therefore did not need to be trained from scratch.

**Camelyon17-WILDS dataset.** The Camelyon17-WILDS dataset consists of histopathology patches from 50 whole slide images collected from 5 different hospitals[29]. These images ($n$: 450, 000) depict lymph node tissue with or without the presence of a tumour. We use the exact same training ($n$: 302, 436), validation ($n$: 33, 560), and test ($n$: 85, 054) splits constructed by the original authors[3]. Notably, the test set contains patches from a hospital whose data was not present in the training set. This setup is therefore meant to reflect the real-world scenario in which models are trained on data from one hospital and deployed on those from another. We chose this dataset as it was purposefully constructed to evaluate the performance of models when presented with data distribution shift.

**Simulated dataset.** We generated a dataset to include a training and held-out set, and data in the wild. To do so, we sampled data from a two-dimensional Gaussian distribution (one for each of the two classes) with diagonal covariance matrices. Specifically, data points from class 1 ($x_1$) and class 2 ($x_2$) were sampled as follows: $x_1 \sim \mathcal{N}([1,1], [0.8, 0.8])$, $x_2 \sim \mathcal{N}([2,2], [0.1, 0.1])$. We assigned 500 and 200 data points to the training and held-out sets. As with the DDI dataset, we did not create a validation set because there was no need to

optimise the model's hyperparameters. As for the data in the wild ($x^W$), these were sampled from different distributions based on the experiment we were conducting. In the out-of-domain setting with and without an imbalance in the number of data points from each class, $x_1^W \sim \mathcal{N}([2, -1], [1,1])$ and $x_2^W \sim \mathcal{N}([3,0], [1,1])$. Data points from a third class were sampled as follows: $x_3^W \sim \mathcal{N}([3, -1], [1,1])$. We assigned 1000 data points to each class for the data in the wild, except for in the label imbalance experiment where we assigned 4000 data points to class 1 and 500 data points to class 2, reflecting an 8 : 1 ratio. For the scenario in which we inject label noise into the held-out dataset, we randomly flip 50% of the labels to the opposite class.

**Multi-domain sentiment dataset.** The multi-domain sentiment dataset consists of reviews of products on Amazon. These products span four different domains from books and electronics to kitchen and DVDs. Each review is associated with either a negative or positive label reflecting the sentiment of the review. In each domain, there are $n = 1000$ reviews reflecting positive and negative sentiment ($n = 2000$ total). When conducting experiments with this dataset, we split the reviews in each domain into training, validation, and test sets using a 60 : 20 : 20 split.

**Flatiron Health ECOG Performance Status (PS).** The nationwide electronic health record (EHR)-derived longitudinal Flatiron Health database, comprises de-identified patient-level structured and unstructured data curated via technology-enabled abstraction, with de-identified data originating from ~ 280 US cancer clinics ( ~ 800 sites of care)[30]. The majority of patients in the database originate from community oncology settings; relative community/academic proportions may vary depending on study cohort. Our dataset, which we term the Flatiron Health ECOG Performance Status database, included 20 disease specific databases available at Flatiron Health as of October 2021 including acute myeloid leukaemia (AML), metastatic breast cancer (mBC), chronic lymphocytic leukaemia (CLL), metastatic colorectal cancer (mCRC), diffuse large B-cell lymphoma (DLBCL), early breast cancer (eBC), endometrial cancer, follicular lymphoma (FL), advanced gastro-esophageal cancer (aGE), hepatocellular carcinoma (HCC), advanced head and neck cancer (aHNC), mantle cell lymphoma (MCL), advanced melanoma (aMel), multiple myeloma (MM), advanced non-small cell lung cancer (aNSCLC), ovarian cancer, metastatic pancreatic cancer, metastatic prostate cancer, metastatic renal-cell carcinoma (mRCC), small cell lung cancer (SCLC), and advanced urothelial cancer. For these patients, the database contains dates of line of therapy (LOT) which is a sequence of anti-neoplastic therapies that a patient receives following the disease cohort inclusion date. The start and end dates of the distinct lines of therapy were captured from both structured and unstructured data sources in the EHR via previously-developed and clinically-informed algorithms. Furthermore, our dataset also contains unstructured clinical notes generated by clinicians that are time-stamped with the visit date (e.g., June 1st, 2017).

**Ethics approval.** The Institutional Review Board of WCG IRB (reference number: IRB00000533) approved of the study protocol prior to study conduct, and included a waiver of informed consent. Patient consent was waived because (a) the research does not involve greater than minimal risk, (b) leverages observational research, which relies on data which was previously collected-as such it is not practicable to conduct the research without the waiver or alteration, and (c) waiving or altering the informed consent will not adversely affect the subjects' rights and welfare. This ethics approval pertains to the use of the Flatiron Health ECOG Performance Status dataset.

**ECOG PS labels.** ECOG PS is a clinical variable that reflects the performance status of an oncology patient. It ranges from 0 (patient has

no limitations in mobility) to 5 (patient deceased)[31] and has been largely used within the context of clinical trials but is also often used by physicians in clinical practice as they make treatment decisions for patients[32,33]. Additionally, in the context of real world evidence, ECOG PS can be used to identify study cohorts of interest (typically those with ECOG PS < 2[34]).

In the Flatiron Health ECOG Performance Status database, the ECOG PS is captured as part of the EHR in either a structured or unstructured form, as outlined next (see Table 1). Structured ECOG PS refers to ECOG values captured in the structured fields of the EHR, such as from drop down lists. In contrast, extracted and abstracted ECOG PS both refer to ECOG values that are captured in unstructured oncologist-generated clinical notes in the EHR. Extracted ECOG PS implies that an NLP symbol-matching (regular expressions) algorithm was used to extract it from clinical notes. This regular expression algorithm was previously developed by researchers at Flatiron Health. Finally, abstracted ECOG PS implies that a human abstractor was able to extract it through manual inspection of clinical notes.

**ECOG task description.** We developed a model which leverages oncology clinical notes to infer a patient's ECOG PS within a window of time (e.g., 30 days) prior to the start of distinct lines of therapy. In oncology research, it is often important to know a patient's ECOG PS at treatment initiation. This allows researchers to investigate the interplay between ECOG PS and different lines of therapy. To achieve this, we consolidated clinical notes within a window of time prior to distinct line of therapy start dates, and combined ECOG PS values (where applicable) using the following strategies.

**Combining clinical notes across time.** We hypothesised that clinical notes up to one month (30 days) before the start date of a line of therapy would contain sufficient information about the performance status of the patient to facilitate the inference of ECOG PS. Given that a patient's ECOG PS can fluctuate over time, retrieving clinical notes further back beyond a month would have potentially introduced superfluous, or even contradictory, information that exacerbated an NLP model's ability to accurately infer ECOG PS. Conversely, exclusively retrieving clinical notes too close to the start date of the LOT might avoid picking up on valuable information further back in time. Based on this intuition, we concatenated all of a patient's clinical notes time-stamped up to and including 30 days before the start date of each of their lines of therapy.

**Combining ECOG PS from multiple sources.** A particular line of therapy for a patient may sometimes be associated with a single ECOG PS (e.g., in structured form) or multiple ECOG PS values (e.g., in extracted and an abstracted form). When multiple sources of ECOG PS scores were available, we consolidated them by only considering the

**Table 1 | Examples of potential sources of the ECOG PS label**

| Scenario | Source of ECOG PS | | |
| --- | --- | --- | --- |
| | Structured | Unstructured | |
| | | Extracted | Abstracted |
| A | list | x | human |
| B | x | regex | human |
| C | x | x | human |
| D | x | x | x |

The ECOG PS clinical variable can be structured; derived from a drop-down list, extracted; through the use of a Flatiron-specific regular expression (regex) algorithm, or abstracted; through the manual inspection of clinical notes by human abstractors.

x indicates the absence of an ECOG PS value from a particular source. The development of an NLP model to infer the ECOG PS is therefore most valuable when none of the the aforementioned sources can produce an ECOG PS value (Scenario D).

largest value (e.g., assign ECOG PS 1 if structured ECOG PS = 0 and abstracted ECOG PS = 1).

Moreover, and without loss of generality, we combined ECOG PS such that [0, 1] map to low ECOG PS and [2, 3, 4] map to high ECOG PS. We chose this binary bucketing since clinical trials in medical oncology typically only include patients with an ECOG PS < 2. The distribution of low and high ECOG PS scores in our study cohort was 82 : 18.

**Description of data sample.** Given the above two consolidation strategies, each sample of data our NLP model was exposed to consisted of (a) concatenated clinical notes for a patient up to 30 days before an LOT and, where available, (b) an ECOG PS label associated with that LOT. Such samples are referred to as labelled. We refer to samples of concatenated clinical notes without an associated ECOG PS label (for details on why, see next section on missingness of ECOG PS) as unlabelled. It is in this scenario where inferring a patient's ECOG PS is most valuable. (see Table 1).

**Missingness of ECOG PS.** There exist a myriad of reasons behind the missingness of the ECOG PS score. For example, in some cases, documenting an ECOG PS score may simply not be a part of a clinician's workflow or may be difficult to document. Alternatively, not documenting an ECOG PS score could be directly related to the qualitative assessment of a patient's health status where its absence could suggest an underlying low ECOG PS value. These reasons, in a majority of settings, imply that the unlabelled clinical notes might follow a distribution that is different from that followed by labelled clinical notes. This discrepancy is also known as covariate shift. As such, we are faced with unlabelled data that exhibit distribution shift. Although our framework's design was motivated by these characteristics, quantifying its utility exclusively on this dataset is challenging due to the absence of ground-truth ECOG PS labels.

**Samples with and without ECOG PS labels.** Our *first study cohort* consists of $n = 117,529$ samples associated with ECOG PS labels. As outlined above, each sample consists of concatenated clinical notes before a particular line of therapy for a patient. When conducting experiments with this cohort, we split the data into training, validation, and test sets using a 70 : 10 : 20 split. This amounted to $n = 81,909$, $n = 11,806$, $n = 23,814$ samples, respectively. Our *second study cohort* consists of $n = 33,618$ samples not associated with ECOG PS labels.

**Description of models**
**Models for image-based datasets.** For the image-based datasets (Stanford DDI and HAM10000), we used two publicly-available models (DeepDerm[15] and HAM10000[16]) that had already been trained on the HAM10000 dataset. We refer readers to the respective studies for details on how these models were trained. In this study, we directly used these models (without retraining) as part of the SUDO experiments. As for the Camelyon17-WILDS dataset, we trained a DenseNet121 model using the default hyperparameters recommended by the original authors[3].

**Models for language-based datasets.** For language-based datasets (Multi-Domain Sentiment and Flatiron Health ECOG PS), we developed a neural network composed of three linear layers which received text as input and returned the probability of it belonging to the positive class, which is positive sentiment for the Multi-Domain Sentiment dataset, and high ECOG PS for the Flatiron Health ECOG Performance Status dataset. An in-depth description of how we pre-processed the input text can be found in the Methods section.

**Pre-processing of text.** We represented text via a bag of words (BoW). This first involved identifying a fixed vocabulary of pairs of words (also known as bigram tokens) present in the training set. After

experimenting with a different number of tokens (e.g., 500, 1,000, 5,000, 10,000), we decided to focus on the 1000 most common tokens as we found that number to provide enough information to learn a generalisable NLP model while not being computationally intensive. The remaining experiments did not result in improved performance. Each document was thus converted into a 1000-dimensional representation (1 dimension for each token) where each dimension reflected the frequency of a particular token in the document. The token mean and standard deviation was calculated across training samples in order to standardise each bigram representation. We found this to result in slightly better performance than settings without input scaling.

**Details of SUDO framework**
SUDO is a framework that helps identify unreliable predictions, select favourable AI models, and assess the algorithmic bias of such systems on data without ground-truth labels. To implement SUDO, we recommend following the steps outlined in the Results (Fig.1b). Here, we provide additional details and intuition about SUDO, mentioning how they align with the previously-outlined steps.

Let us assume we have a probabilistic model that returns a single value reflecting the probability, $s$, that an input belongs to the positive class (e.g., high ECOG PS in the Flatiron Health ECOG PS dataset). We can generate a distribution of such probability values for all data points and discretize the distribution in probability intervals (Fig. 1b, Step 1 and Step 2).

**Sample data points.** From each probability interval $s \in (s_1, s_2]$ where $s_1 < s_2$, we sampled a subset of the data points (Fig. 1b, Step 3). To avoid sampling more data points from one probability interval than from another, and potentially affecting the reliability of estimates across intervals, we fixed the number of data points, $m$, to sample from each interval. Specifically, $m$ was chosen based on the lowest number of data points within an interval, across all probability intervals. For example, if the interval $s \in (0.4, 0.5]$ contains the lowest number of data points (e.g., 50), then we sample $m = 50$ data points from each interval. This also ensures that we sample data points without replacement to avoid a single data point from appearing multiple times in our experiments and biasing our results. Next, we assigned these sampled data points a temporary label, also known as a pseudo-label, hypothesizing that they belong to a certain class (e.g., class 0).

**Train classifier.** We then trained a classifier, $g_\phi$, to distinguish between such newly-labelled data points and data points with a ground-truth label from the opposite class (e.g., class 1) (Fig. 1b, Step 4). It is worthwhile to mention that this classifier need not be the exact same model as the one originally used to perform inference (i.e., $g_\phi \neq f_\theta$). The prime desiderata of the classifier are that it (i) is sufficiently expressive such that it can discriminate between positive and negative examples and (ii) can ingest the input data. Such a modular approach, where one model is used for the original inference (Fig. 1b, Step 1) and another is used to distinguish between positive and negative examples ((Fig. 1b, Step 4) is less restrictive for researchers and can obviate the need to (re)train potentially computationally expensive inference models. This line of argument also extends to settings with different data modalities (e.g., images, time-series, etc.), and, as such, makes SUDO agnostic to the modality of the data used for training and evaluating the model.

**Evaluate classifier.** After training $g_\phi$, we evaluated it on a held-out set of data comprising data points with ground-truth labels (from both class 0 and class 1) (Fig. 1b, Step 5). The intuition here is that a classifier which can successfully distinguish between these two classes, by performing well on the held-out set of data, is indicative that the training data and the corresponding ground-truth labels are relatively reliable. Since the data points from class 1 are known to be correct (due to our

use of ground-truth labels), then a highly-performing classifier would suggest that the class 0 pseudo-labels of the remaining data points are likely to be correct. In short, this step quantifies how plausible it is that the sampled unlabelled data points belong to class 0.

As presented, this approach determines how plausible it is that the sampled set of data points in some probability interval, $s \in (s_1, s_2]$, belongs to class 0. It is entirely possible, however, that a fraction of these sampled data points belong to the opposite class (e.g., class 1). We refer to this mixture of data points from each class as class contamination. We hypothesised (and indeed showed) that the degree of this class contamination increases as the probability output, $s$, by an AI system steers away from the extremes ($s \approx 0$ and $s \approx 1$). To quantify the extent of this contamination, however, we also had to determine how plausible it was that the sampled set of data points belong to class 1, as we outline next.

**Cycle through the pseudo-labels.** We repeated the above approach (Fig. 1b, Steps 3, 4, and 5) however with two distinct changes. First, we pseudo-labelled the sampled data points with class 1 (instead of class 0). In doing so, we are hypothesizing that these data points belong to class 1. Second, we trained a classifier to distinguish between such newly-labelled data points and data points with ground-truth labels from class 0.

When experimenting with the distinct pseudo-labels, we always sample the same set of data points, as enforced by a random seed. Doing so ensures that any difference in the predictive performance of the learned classifiers, $g_\phi$, is less attributable to differences in the sampled data points and, instead, more attributable to the pseudo-labels that we have assigned. Moreover, to ensure that our approach is not constrained by a particular subset of sampled data points, we repeat this entire process multiple ($k = 5$) times, each time sampling a different subset of data points from each probability interval, as enforced by a random seed (e.g., 0 to 4 inclusive).

**Derive the pseudo-label discrepancy.** The discrepancy between, and ranking of, the classifier performances above is indicative of data points that are more likely to belong to one class than another. Concretely, if the classifier, $g_\phi$, achieves higher performance when presented with sampled data points that are pseudo-labelled as class 0 than as class 1, then the set of pseudo-labelled data points are more likely to belong to class 0. We refer to this discrepancy in performance under different scenarios of pseudo-labels as the pseudo-label discrepancy, or SUDO.

## Implementation details of SUDO experiments

SUDO involves selecting several hyperparameters. These can include the granularity and number of probability intervals, the number of data points to sample from each probability interval, the number of times to repeat the experiment, and the type of classifier to use. We offer guidelines on how to select these hyperparameters in the Results.

**Stanford DDI dataset.** For the DeepDerm model (Fig. 2a), we selected ten mutually-exclusive and equally-sized probability intervals in the range $0 < s < 1$, and sampled 10 data points from each probability interval. For the HAM10000 model (Fig. 2b), we selected ten mutually-exclusive and equally-sized probability intervals in the range $0 < s < 0.5$, and sampled 50 data points from each probability interval. In the latter setting, we chose a smaller probability range and more granular probability intervals to account for the high concentration of data points as $s \to 0$.

To amortize the cost of training classifiers as part of the SUDO experiments, we extracted image representations offline (before the start of the experiments) and stored them for later retrieval. To capture a more representative subset of the data points and obtain a better estimate of the performance of these classifiers, we repeated

these experiments 5 times for each probability interval and pseudo-label. To accelerate the experiments, we used a lightweight classifier such as a logistic regression, discovering that more complex models simply increased the training overhead without altering the findings. Unless otherwise noted, we adopted this strategy for all experiments.

**Camelyon17-WILDS dataset.** We selected eleven mutually-exclusive and equally-sized probability intervals in the range $0.10 < s < 0.75$ which was chosen based on where the AI-based probability values were concentrated. We sampled 1000 data points from each probability interval. To remain consistent with the other experiments in this study, we used the provided in-distribution validation set as the held-out set (Fig. 1, Step 5). As with the Stanford DDI dataset, to minimise the cost of conducting the SUDO experiments, we first extracted and stored the image representations of the histopathology patches using the trained DenseNet121 model. We otherwise followed the same approach as that mentioned above.

**Multi-domain sentiment dataset.** We also selected ten mutually-exclusive and equally-sized probability intervals in the range $0 < s < 1$, and sampled 50 and 10 data points from each probability interval when dealing with a network that was not overconfident and one that was trained to be overconfident. We sampled fewer data points in the latter setting because prediction probability values were concentrated at the extreme ends of the probability range ($s \to 0$ and $s \to 1$), leaving fewer data points in the remaining probability intervals. As demonstrated in the Results, SUDO can deal with such data-scarce settings.

**Simulated dataset.** We selected ten mutually-exclusive and equally-sized probability intervals in the range $0 < s < 1$, and sampled 50 data points from each probability interval.

**Flatiron Health ECOG PS dataset.** On the Flatiron Health ECOG Performance Status dataset with ECOG PS labels, we sampled 200 data points from each probability interval. This value was chosen to capture a sufficient number of predictions from each probability interval without having to sample with replacement. On the Flatiron Health ECOG Performance Status dataset without ECOG PS labels (data in the wild), we sampled 500 data points from each interval in the range $(0, 0.45]$ and 100 data points from each interval in the range $(0.45, 1]$. This was due to the skewed distribution of the probability values generated in such a setting (see Fig. 4b).

## Implementation details of algorithmic bias experiment

To demonstrate that SUDO can help assess algorithmic bias on data without ground-truth annotations, we conducted experiments on the Stanford DDI dataset because those images could be categorised based on a patient's skin tone (Fitzpatrick I-II, III-IV, V-VI). As such, we would be able to assess the bias of the pre-trained AI systems against particular skin tones.

We followed the same steps to implement SUDO (see Fig. 1b). The main difference is that we first stratified the data points according to skin tone. Based on the bias reported in a recent study[14], we focused on skin tone I-II and V-VI. Although such stratification can be done within each probability interval, after having observed the the HAM10000 model's prediction probability values (Fig. 2b), we considered a single probability interval $0 < s < 0.20$ where data points would be classified as negative (benign lesions). We sampled 200 data points from this probability interval for each group (I-II and V-VI) to conduct the SUDO experiments and calculated the AUC of the subsequently-learned classifiers.

## Applications of SUDO

SUDO can help with identifying unreliable clinical predictions, selecting favourable AI systems, and assessing the bias of such systems, as outlined next.

**Identifying unreliable AI-based predictions.** Identifying unreliable AI-based predictions, those whose assigned label may be incorrect, is critical for avoiding the propagation of error through downstream research analyses. SUDO allows for this as it provides an estimate of the degree of class contamination for data points whose corresponding AI-based output probabilities are in some probability interval. Specifically, a $\downarrow|D|$ (small difference in classifier performance across pseudo-label settings) implies $\uparrow$ class contamination. As such, by focusing on probability intervals with $|D| < \tau$ where $\tau$ is some predefined cutoff, one can now identify unreliable AI-based predictions. As we will show, there is an even greater need to identify such contamination when dealing with over-confident AI systems.

**Selecting AI systems.** An AI system is often chosen based on its reported performance on a held-out set of data. We define a favourable AI system as that which performs best on a held-out set of data compared to a handful of models. The ultimate goal is to deploy the favourable model on data in the wild. However, with data in the wild exhibiting a distribution shift and lacking ground-truth labels, it is unknown what the performance of the chosen AI system would be on the data in the wild, thereby making it difficult to assess whether it actually is favourable for achieving its goal.

**Assessing algorithmic bias.** Assessing algorithmic bias is critical for ensuring the ethical deployment of AI systems. A common approach to quantify such bias is through a difference in AI system performance across groups of patients (e.g., those in different gender groups)[35]. The vast majority of these approaches, however, requires ground-truth labels which are absent from data points in the wild thereby making an assessment of bias out-of-reach. However, SUDO, by producing a reliable proxy for model performance, allows for this capability.

### Implementation details of survival analysis
We assessed real world overall survival defined as time from the start of first line of therapy (LOT = 1) to death[36]. If death was not observed by the study end date (October 2021), patients were censored at the date of a patient's latest clinical visit. We estimated survival using the Kaplan Meier method and used lifelines package to conduct our analysis[37]. To avoid confounding due to lines of therapy, we conducted all survival analyses for patients receiving their first line of therapy only (i.e., LOT = 1). No other adjustments were made.

**Steps to generate Fig. 4e.** We first filtered our data samples with known ground-truth ECOG PS labels ($n = 117,529$) to only consider those tagged with the first line of therapy (LOT = 1). Using these samples, we conducted two survival analyses: one with data samples for patients with a low ECOG PS label and another for patients with a high ECOG PS label.

**Steps to generate Fig. 4f.** We filtered our data samples in the data in the wild ($n = 33,618$) to only consider those tagged with the first line of therapy (LOT = 1). However, since these data samples were not associated with a ground-truth ECOG PS label, we split them into three groups based on a chosen threshold on the pseudo-label discrepancy presented in Fig. 4d. Using the intuition that a higher absolute pseudo-label discrepancy is indicative of more reliable predictions, we chose three probability intervals to reflect the three distinct patient cohorts: low ECOG PS group ($0 < s \leq 0.2$), high ECOG PS group ($0.5 \leq s < 1.0$), and an uncertain ECOG PS group ($0.2 < s < 0.5$). We subsequently conducted a survival analysis using the data samples in each group.

**Steps to generate Fig. 4h.** We conducted multiple survival analyses. Each analysis was performed as described above and for the subset of patients whose associated AI-based predictions fell in a probability interval (e.g., $0 < s \leq 0.05$, $0.05 < s < 0.10$, etc.). Since there were 14 probability intervals in total, we performed 14 survival analyses and calculated the median survival time in each analysis. This allowed us to correlate the median survival time, per probability interval, to the derived pseudo-label discrepancy.

### Producing reliability-completeness curves
The completeness of a variable (the proportion of missing values that are inferred) is equally important as the reliability of the predictions that are being made by a model. However, these two goals of data completeness and data reliability are typically at odds with one another. Quantifying this trade-off confers a twofold benefit. It allows researchers to identify the level of reliability that would be expected when striving for a certain level of data completeness. Moreover, it allows for model selection, where preferred models are those that achieve a higher degree of reliability for the same level of completeness. To quantify this trade-off, we needed to quantify the reliability of predictions without ground-truth labels and their completeness. We outline how to do so next.

**Quantify reliability.** SUDO reflects the degree of class contamination within a probability interval. The higher the absolute value of SUDO, the lower the degree of class contamination. Given a set of low probability thresholds, $\alpha \in A$, and high probability thresholds, $\beta \in B$, we can make predictions $\hat{y}$ of the following form,

$$\hat{y} = \begin{cases} 0, & s \leq \max(A) \\ 1, & s \geq \min(B) \end{cases} \quad (1)$$

To calculate the reliability $R_{A,B}$ of such predictions, we could average the absolute values of SUDO for the set of probability thresholds ($A$, $B$),

$$R_{A,B} = \frac{1}{2 \cdot |A||B|} \sum_{\alpha \in A, \beta \in B} |\text{SUDO}(\alpha)| + |\text{SUDO}(\beta)| \quad (2)$$

**Quantify completeness.** By identifying the maximum probability threshold in the set, $A$, and the minimum probability threshold in the set, $B$, the completeness, $C_{A,B} \in [0, 1]$, can be defined as the fraction of all data points in the wild, M, that falls within this range of probabilities,

$$C_{A,B} = \sum_{j=1}^{M} \mathbb{1}[s_j \leq \max(A) \text{ or } s_j \geq \min(B)] \quad (3)$$

**Generate reliability-completeness curve.** After iterating over $K$ sets of $A$ and $B$, we can populate the reliability-completeness (RC) curve for a particular model of interest (see Fig. 2e). From this curve, we derive the area under the reliability-completeness curve, or the AURCC $\in [0, 1]$.

$$\text{AURCC} = \frac{1}{2K} \sum_{k=1}^{K} \frac{R_{A,B}(k) + R_{A,B}(k+1)}{\Delta C_{A,B}} \quad (4)$$

Whereas the area under the receiver operating characteristic curve (AUROC) summarises the performance of a model when deployed on labelled instances, the AURCC does so on unlabelled data points. Given this capability, the AURCC can also be used to compare the performance of different models.

### Reporting summary
Further information on research design is available in the Nature Portfolio Reporting Summary linked to this article.

## Data availability
The Stanford diverse dermatology images (DDI) dataset is available at https://ddi-dataset.github.io/index.html#access. The Camelyon17-WILDS dataset is available at https://wilds.stanford.edu/get_started/.

The Multi-Domain Sentiment dataset is available at https://www.cs.jhu.edu/m̃dredze/datasets/sentiment/index2.html. All publicly-available datasets were used as permitted, exclusively for research purposes. The Flatiron Health ECOG PS dataset is available under restricted access due to patient privacy. Requests for data sharing by license or by permission for the specific purpose of replicating results in this manuscript can be submitted to PublicationsDataAccess@flatiron.com. A response will be provided within a week of receiving the request. The data for generating the figures in this study are in the Source Data file. Source data are provided with this paper.

## Code availability

Our code is publicly available at https://github.com/flatironhealth/SUDO.

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

## Acknowledgements
We are grateful to Alex Rich, Selen Bozkurt, and Emily Castellanos for providing feedback at various stages of the manuscript. We also thank Guy Amster for facilitating the internal review of the manuscript.

## Author contributions
D.K. and N.A. contributed to the conception of the study. D.K. contributed to the study design, conducted the experiments, and wrote the manuscript. C.J. informed the survival analysis study and provided feedback on the manuscript. A.C. informed the clinical impact and provided feedback on the manuscript.

## Competing interests
A.C., C.J., and N.A. report employment with Flatiron Health Inc., which is an independent member of the Roche Group, and stock ownership in Roche. The remaining authors declare no competing interests. A patent has been filed for the methods described in this study.
