## [Peer Review File · Nature Communications]

Reviewers' Comments:

Reviewer #1:

Remarks to the Author:

The authors aim to tackle an important problem that clinical ML commonly faces, which is estimating the likely performance and failure modes of models at deployment when they are probed with real-world examples.

The manuscript clearly motivates the problem and explains the experimental setting in which the proposed approach, SUDO, is evaluated.

I would like to share a few thoughts on why I believe the paper requires more rigorous experimentation and methodological explanations to justify claims such as "it enables the evaluation of clinical ML without ground-truth annotations."

1) The experiments were conducted on two datasets, one of which (Multi-Domain Sentiment) is a general domain dataset that does not relate to clinical data. Further, the second dataset is constructed based on assumptions that do not guarantee the presence of "in the wild" or "distribution-shift" scenarios. The authors explicitly acknowledge this in the text, stating "we hypothesize that this could result in a shift in the data distribution." From this perspective, the experimental evidence does not seem to be sufficient to make such claims. I would recommend using publicly available domain-shift datasets to validate the proposed solution, without being limited to NLP-only datasets. Additionally, the validity of the identified problems may not persist or be applicable for the language modality when large language models (LLMs) are used; thus, validating the approach on other data modalities could broaden the scope of the proposed algorithm.

2) The proposed solution (SUDO) is explained in plain language as an algorithm; however, I find there is a lack of motivation and critical reasoning regarding potential corner cases where this algorithm could fail. For instance, the authors mention that deep neural nets can make class posterior predictions with very high confidence and still fail, which could render the inference probability estimates collected on "in-the-wild" samples (external) less meaningful. For example, what would happen if the initial model assigns "class-0" to all external samples as pseudo-labels in a systematic manner? I would expect the subsequent two models to differ in terms of their predictions; however, this does not necessarily mean that the initial classification results were correct. Without reliable confidence estimates and the injection of any additional information into the system, it is difficult to comprehend how such a system would recognize what it actually doesn't know. Therefore, the study requires more theoretical justification, along with additional experimental analysis, to support the claims made.

Reviewer #2:

Remarks to the Author:

I thank the authors for addressing a pressing and relatively unaddressed problem within machine learning (ML) in medicine. This paper introduces a novel framework, entitled pseudo-label discrepancy (SUDO), with 3 primary use cases for an artificial intelligence (AI) system deployed on data without ground-truth labels and potential distributional shift: (i) identification of unreliable AI-based predictions, (ii) selection of favorable AI systems for achieving some goal, and (iii) assessment of the algorithmic bias. The authors provide an extensive evaluation of their framework using two real-world datasets, the Multi-Domain Sentiment dataset and the Flatiron Health Eastern Cooperative Oncology Group Performance Status (ECOG PS) dataset.

I have significant concerns about the proposed methodology and its relationship to existing methods. I therefore focus my comments on the methods rather than more detailed questions regarding the analyses.

Major Comments

(1) Although the authors discuss related literature in identifying unreliable AI predictions in the introduction and reverse testing and validation in the conclusion, there are important gaps in the literature review with respect to evaluating model accuracy with scarce or no labeled data. For instance, within ML there has been some work in classification accuracy and fairness assessment in the positive-unlabeled and semi-supervised settings (e.g., [1,2]). Within statistics, there is a vast literature on predictive performance evaluation in the presence of missing labeled data, with early methods taking a missing data viewpoint (e.g., [3,4]). More recent proposals target challenges relevant to the analysis of electronic health records data, including a lack of labeled data and distribution shift (e.g., [5–8]). Methods that operate without labeled data have also been previously considered (e.g., [9]). It is necessary to draw connections with these works and characterize the contribution of this paper in light of these developments.

(2) The authors provide intuition for the mechanics of SUDO, but no technical arguments are provided. For example, under what conditions will SUDO provide consistent results with the “oracle method” based on evaluation with fully labeled data in the wild? These details need to be fleshed out given the importance of obtaining reliable conclusions for the three applications of interest. Simulation studies considering a broader range of settings in which the data generating mechanism is known would also assist in providing a more comprehensive evaluation of SUDO.

(3) Methods for statistical inference are neither provided nor discussed. For example, one would need procedures for statistical testing and interval estimation to evaluate algorithmic bias and to compare the area under the reliability completeness curve for two candidate models.

(4) I found the description of SUDO difficult to follow. I was initially going back and forth between Figure 1, the “Mechanics of SUDO” section, and the Methods section. More concretely, the description in the “Mechanics of SUDO” section and the presentation in Figure 1 are somewhat incongruent. The description of the pseudo-label discrepancy is also lacking detail. There is no explicit statement of (i) how the pseudo-label discrepancy is used for the three applications of identifying unreliable AI predictions, selecting AI systems, and assessing fairness of AI predictions and (ii) why SUDO is agnostic to the choice of metric.

(5) The proposed usage of the training data, held-out data, and data in the wild sets would benefit from more justification. The setting under consideration has similarities to that in [6]. Can the authors comment on the advantage of their approach and how it relates to this approach?

(6) SUDO relies on several hyperparameter and modeling choices, including the discretion into quantiles, the number of samples from each quantile, the choice of classifier, choice of discrepancy metric and threshold, etc. Can the authors provide more discussion and/or evaluation of the sensitivity of their method to these choices? What specific guidance can be provided to practitioners?

(7) The MultiSentiment dataset is used to illustrate the utility of SUDO. While I agree with the authors that this dataset is a good testing ground, the paper is motivated by the application of AI in medicine. It would be preferable for this analysis to be supplemental and for a publicly available medical dataset to be presented in the main text. For example, the MIMIC dataset (or potentially the MIMIC phenotype annotation dataset) is one option and well-known by the ML for health community.

(8) In the Multi-Domain Sentiment analysis, why are fairness evaluations not considered? Is information on protected attributes not available? If so, can a simulated variable be created?

(9) In the first Flatiron ECOG PS analysis, there does not appear to be any bias across groups defined by gender. Can the authors provide evaluation for a protected attribute where bias is present to illustrate SUDO performs well in this setting?

(10) In the survival analysis of the Flatiron ECOG PS dataset, there is a well known relationship between mortality and ECOG PS. How would one evaluate SUDO in a setting where no such knowledge exists? This is briefly mentioned in the limitations discussion at the end of the paper, but I would suggest more detail on this point.

(11) At a more fundamental level, any method for evaluating an AI method that does not require any gold-standard labeling of the data in the wild will inevitably rely on more modeling assumptions and in turn has the potential to lead to distorted conclusions. Can the authors add a discussion on the ethical and practical considerations related to the trade-off between more time-consuming traditional evaluation vs. the proposed approach?

Minor Comments

(1) How much variability is there in the quality of the various ECOG PS labels presented in Table 1? What does "I" mean in this table?

(2) A footnote states that SUDO "trivially extends to multi-class problems." I suggest this be detailed in the supplement.

(3) Fairness evaluation is only considered for a binary attribute. A short discussion of evaluation across more than 2 groups would be useful given numerous metrics have been proposed in the literature.

(4) Is the training set in Figure 1 the same data used to train the AI system? Would this lead to bias in the three applications?

(5) More discussion of how to interpret and assess Figures 3g and 3h would be helpful.

(6) Small typos in the text (e.g., "in" is needed after "confidence" in the first sentence of the "Validating SUDO-guided predictions with a survival analysis" section).

(7) If I understand correctly, it also seems the point estimates for the median survival times for the two groups from SUDO vs. the ground truth are quite different. Is this expected?

(8) While I do agree that unreliable predictions can be flagged for further review, I disagree with the statement that "excluding such unreliable predictions from analyses improves the integrity of research findings." Excluding such predictions has the potential to bias results. Ideally, analysis should be conducted after further interrogation of unreliable predictions so they may be included in analysis.

(9) Will any code for SUDO be made publicly available?

References

1. Claesen M, Davis J, De Smet F, De Moor B. Assessing binary classifiers using only positive and unlabeled data. arXiv [stat.ML]. 2015. Available: <http://arxiv.org/abs/1504.06837>
2. Ji D, Smyth P, Steyvers M. Can I trust my fairness metric? assessing fairness with unlabeled data and bayesian inference. Adv Neural Inf Process Syst. 2020;33: 18600–18612.
3. Fluss R, Reiser B, Faraggi D, Rotnitzky A. Estimation of the ROC curve under verification bias. Biom J. 2009;51: 475–490.
4. Rotnitzky A, Faraggi D, Schisterman E. Doubly Robust Estimation of the Area Under the Receiver-Operating Characteristic Curve in the Presence of Verification Bias. J Am Stat Assoc. 2006;101: 1276–1288.
5. Gronsbell JL, Cai T. Semi-supervised approaches to efficient evaluation of model prediction performance. Journal of the Royal Statistical Society: Series B (Statistical Methodology). 2018. pp. 579–594. doi:10.1111/rssb.12264
6. Wang L, Wang X, Liao KP, Cai T. Semi-supervised Transfer Learning for Evaluation of Model Classification Performance. arXiv [stat.ME]. 2022. Available: <http://arxiv.org/abs/2208.07927>
7. Zhou D, Liu M, Li M, Cai T. Doubly Robust Augmented Model Accuracy Transfer Inference with High Dimensional Features. arXiv [stat.ME]. 2022. Available: <http://arxiv.org/abs/2208.05134>
8. Gronsbell J, Liu M, Tian L, Cai T. Efficient Evaluation of Prediction Rules in Semi-Supervised Settings under Stratified Sampling. J R Stat Soc Series B Stat Methodol. 2022;84: 1353–1391.
9. Joseph L, Gyorkos TW, Coupal L. Bayesian estimation of disease prevalence and the parameters

of diagnostic tests in the absence of a gold standard. *Am J Epidemiol.* 1995;141: 263–272.

We would like to thank the reviewers for taking the time and effort to read our manuscript and for providing us with valuable feedback.

In addition to addressing each of your comments, we have grouped those with a common theme and addressed them first. The two high level themes are:

- 1) additional experimentation on public datasets
- 2) additional experimentation to stress-test SUDO

THEMES

Theme 1 – experimentation on publicly-available datasets with distribution shift

To demonstrate the applicability of SUDO to datasets of different modalities with a known distribution shift, we have now conducted additional experiments on the Stanford diverse dermatology images dataset. Specifically, the Results section now contains the following subsections (alongside Figure 2, page 4):

- *SUDO correlates with model performance on Stanford dermatology images dataset* – we present evidence that SUDO continues to correlate well with model accuracy when used with two different convolutional neural networks deployed on a dataset of images. Such a finding points to the applicability of SUDO to datasets of different modalities. Results → SUDO correlates with model performance on Stanford dermatology images dataset (page 3)

SUDO correlates with model performance on Stanford diverse dermatology images dataset

We used SUDO to evaluate predictions made on the Stanford diverse dermatology image (DDI) data¹³ ($n : 656$) (see Description of datasets). We purposefully chose two AI models (DeepDerm¹⁴ and HAM10000¹⁵) that were performant on their respective data (AUC = 0.88 and 0.92) and whose performance degraded drastically when deployed on the DDI data (AUC = 0.56 and 0.67), suggesting the presence of distribution shift.

Confirming previous findings, we found that these models struggle to distinguish between benign (negative) and malignant (positive) lesions in images. This is evident by the lack of separability of the AI-based probabilities corresponding to the ground-truth negative and positive classes (Fig. 2a and b). We set out to determine whether SUDO can quantify this class contamination (without ground-truth labels). By following the steps of SUDO (see Overview of the SUDO framework), we found that it correlates ($\rho = -0.84$ and -0.76 for DeepDerm and HAM10000, respectively) with the proportion of positive instances in each of the chosen probability intervals (Fig. 2c and d). Such a finding, which holds even if we use a different evaluation metric for SUDO (Supplementary Fig. 5), suggests that SUDO can act as a reliable proxy for the accuracy of predictions. Notably, this ability holds irrespective of the underlying performance of the AI model being evaluated, as evidenced by the high correlation values for the two models with different performance metrics (AUC = 0.56 and 0.67).

- *SUDO informs model selection with Stanford diverse dermatology images dataset* – we introduce our reliability-completeness curve early on in the manuscript in order to clarify the potential application areas of SUDO. We show that SUDO can correctly rank models even without ground-truth annotations, and that this is consistent with previously-reported findings of the performance of such models. **Results → SUDO informs model selection with Stanford diverse dermatology images dataset (page 3)**

SUDO informs model selection with Stanford diverse dermatology images dataset

As a proxy for the accuracy of AI predictions, SUDO can identify two tiers of predictions: those which are sufficiently reliable for downstream analyses and others which are unreliable and might require further inspection by a human expert. This creates a trade-off between the completeness of AI predictions (i.e., are *all* predictions included in downstream analyses?) and the reliability of such predictions. Ideally, both of these elements are maximized for AI predictions. We leverage this intuition and introduce the reliability-completeness curve as a way of rank ordering models when ground-truth annotations are unavailable (Fig. 2e, see Producing reliability-completeness curve for details).

We found that the ordering of the performance of the models is consistent with that presented in previous studies¹¹. Specifically, HAM10000 and DeepDerm achieve an area under the reliability-completeness curve (AURCC = 0.864 and 0.621, respectively) and, with ground-truth annotations, these models achieve (AUC = 0.67 and 0.56). We note that the emphasis here is on the relative ordering of models and not on their absolute performance. These consistent findings suggest that SUDO can help inform model selection on data in the wild without annotations.

- *SUDO helps assess algorithmic bias without ground-truth annotations* – we also present evidence early on in the manuscript that SUDO can help assess algorithmic bias in order to clarify another potential application area of SUDO. We show that SUDO, when stratified across patient groups with different skin tones, can detect a bias in favour of patients with a lighter skin tone (Fitzpatrick scale I-II), and which is consistent with previously-reported findings. **Results → SUDO helps assess algorithmic bias without ground-truth annotations (page 3-4)**

SUDO helps assess algorithmic bias without ground-truth annotations

Algorithmic bias often manifests as a discrepancy in model performance across two protected groups (e.g., male and female patients). Traditionally, this would involve comparing AI predictions to ground-truth labels. Viewing SUDO as a proxy for model performance, we hypothesized that it can help assess such bias even without ground-truth labels. We tested this hypothesis on the Stanford DDI dataset by stratifying the AI predictions according to the skin tone of the patients (Fitzpatrick scale I-II vs. V-VI) and implementing SUDO for each of these stratified groups. A difference in the resultant SUDO values would indicate a higher degree of class contamination (and therefore poorer performance) for one group over another. Having found that $SUDO_{AUC} = 0.60$ and 0.58 for

the two groups, respectively, we supported the validity of this bias by using the ground-truth labels to calculate the negative predictive value of the predictions (NPV). With NPV = 0.83 and 0.78, respectively, we found that both SUDO and the traditional approach (with ground-truth labels) identified a bias in favour of patients with a Fitzpatrick scale of I-II; the main distinction is that SUDO did not require ground-truth labels for the dataset being evaluated. Moreover, these bias findings are consistent with those reported in a previous study¹³.

Figure 2. SUDO is a reliable proxy for model performance on the Stanford diverse dermatology image dataset. Two models (left column: DeepDerm, right column: HAM10000) are pre-trained on the HAM10000 dataset and deployed on the entire Stanford DDI dataset. (a-b) Distribution of the prediction probability values produced by the two models colour-coded based on the ground-truth label (negative vs. positive) of the data points. (c-d) Correlation of SUDO with the proportion of positive data points in each probability interval. Results are shown for ten mutually-exclusive probability intervals that span the range [0, 1]. A strong correlation indicates that SUDO can be used to identify unreliable predictions. (e) Reliability-completeness curves of the two models, where the area under the reliability-completeness curve (AURCC) can inform the selection of an AI system without ground-truth annotations.

Theme 2 – experimentation to “stress-test” SUDO and offer guidance on its use

To stress test SUDO, we have now conducted additional experiments on simulated data whereby we vary the characteristics of the data in the wild. Specifically, we vary the data in the wild to exhibit (a) distribution shift, (b) distribution shift with an imbalance in the number of data points per class, (c)

distribution shift with the presence of a third-and-unseen class, and (d) distribution shift where the held-out set contains label noise.

- *Exploring the limits of SUDO with simulated data* – we show that SUDO continues to correlate well with model performance in the above scenarios (except for when label noise is present and the relationship between the data in the wild distributions is drastically changed), demonstrating its robust behaviour. These findings are presented in Supplementary Figures 2 and 3 and summarized in Results → Exploring the limits of SUDO with simulated data (page 5).

Exploring the limits of SUDO with simulated data

To shed light on the scenarios in which SUDO remains useful, we conducted experiments on simulated data that we can finely control (see Description of datasets). Specifically, we varied the data in the wild to encompass distribution shift (a) with the same two classes observed during training, (b) with a severe imbalance (8 : 1) in the number of data points from each class, and (c) alongside data points from a third and never-seen-before class. As SUDO is dependent on the evaluation of classifiers on held-out data (see Fig. 1, Step 5), we also experimented with injecting label noise into such data.

We found that SUDO continues to strongly correlate with model performance, even in the presence of a third class ($|\rho| > 0.87$, Supplementary Fig. 2). This is not surprising as SUDO is designed to simply quantify class contamination in each probability interval, regardless of the data points contributing to that contamination. However, we did find that SUDO requires held-out data to exhibit minimal label noise, where $\rho = 0.99 \rightarrow 0.33$ upon randomly flipping 50% of the labels in the held-out data to the opposite class. We also found that drastically changing the relationship between class-specific distributions of data points in the wild can disrupt the utility of SUDO (Supplementary Fig. 3).

Exploring the limits of SUDO on simulated data

We implemented SUDO on simulated data in attempt to explore the limits of SUDO's applicability. We first experimented with multiple scenarios in which the data in the wild is varied by, for example, introducing an imbalance in the number of data points from each class (Supplementary Fig. 2b) or introducing data points from a third-and-unseen class (Supplementary Fig. 2c). The details of how we sampled the data can be found in the Methods section and the findings are summarized in the Results section.

Supplementary Figure 2. SUDO continues to act as a reliable proxy for model performance under various scenarios. We implement SUDO on simulated data where the data in the wild exhibit (left column) distribution shift, (middle column) distribution shift with an imbalance in the number of data points from each class, and (right column) distribution shift with the presence of a third class. **(a-c)** Scatter plot of the data points in the training set (light shade) and in the wild (dark shade). **(d-f)** Distribution of the prediction probability values of models deployed on the data in the wild, colour-coded according to the ground-truth classes. **(g-i)** Correlation between the SUDO values and the proportion of positive instances in each of the probability intervals.

We conducted additional experiments to further probe the limits of SUDO's applicability. In particular, we explored the scenario in which the distribution of data points in the wild from one class remains fixed whereas the distribution of data points from the opposite class varied. In the baseline scenario (Supplementary Fig. 3a), we previously demonstrated that SUDO strongly correlates with the proportion of positive instances in each probability interval ($\rho = -0.99$). Note that the negative direction of this relationship is expected based on how we defined SUDO (see Supplementary Figs. 2g-i for additional evidence).

We then held fixed the distribution of the data points in the wild from class 0, and exclusively varied the mean of the distribution of the data points in the wild from class 1 (Supplementary Figs. 3b-c). The latter distribution was shifted to overlap with the former distribution (Supplementary Fig. 3b) and ultimately to swap its ordering (Supplementary Fig. 3c). We quantified this change as the distance of the mean from that in the baseline scenario (Supplementary Fig. 3a). We illustrate the correlation of SUDO with the proportion of positive instances as a function of these shifted distributions (Supplementary Fig. 3d). We found that SUDO no longer correlates ($\rho = 0$) with the proportion of positive instances when data points in the wild from opposite classes share similar features and are difficult to distinguish from one another. We also found that SUDO's relationship with the proportion of positive instances was inverted when the ordering of the distributions of data in the wild differs from that of training data. For example, $\rho \approx 0.65$ at a distribution change of $3\sqrt{2}$ (Supplementary Fig. 3d). Based on how we defined SUDO, this finding suggests that SUDO would erroneously indicate that the majority of data points in a probability interval belonged to class 0 when in reality they belonged to class 1. Although this scenario may be somewhat fictitious, it helps identify when SUDO should not be depended on.

Supplementary Figure 3. SUDO can sometimes fail to act as a reliable proxy for model performance. We implement SUDO on simulated data where we hold fixed the distribution of data in the wild from class 0, and vary the distribution of data from the class 1. **(a-c)** Scatter plot of the data points in the training set (light shade) and in the wild (dark shade). **(d)** Correlation between the SUDO values and the proportion of positive instances as a function of the change in the mean of the distribution of data points from class 1. We show that extreme changes in that distribution can disrupt the reliability of SUDO as a proxy for model performance.

We have since added content, beyond that outlined in the above themes, to the manuscript. These include:

- 1) Experiments to measure the sensitivity of SUDO to various hyperparameters
- 2) Experiments to show that SUDO is agnostic to the metric used to evaluate classifiers
- 3) Improved description of the SUDO framework
- 4) Improved description of related work
- 5) Practical guidelines for using SUDO

ADDITIONS

Addition 1 – Experiments to measure the sensitivity of SUDO to various hyperparameters

We implement SUDO on the Flatiron Health ECOG PS dataset while varying the number of data points sampled from each probability interval, the type of classifier used to distinguish between pseudo and ground-truth labelled data points, and the amount of label noise in the held-out data being evaluated on.

We show that SUDO continues to correlate well with model performance even when we sample as few as 10 data points from each probability interval, suggesting that it can be applied in data-scarce settings. While we also show that SUDO is agnostic to the type of classifier used (e.g., logistic regression vs. random forest), it is surely affected by the presence of label noise in the held-out set of data. We present the results in Supplementary Material → Supplementary Note 1 → SUDO is insensitive to various hyperparameters, and summarize the findings in Results → Sensitivity analysis of SUDO's hyperparameters (page 5). We ultimately use these findings to inform the practical guidelines we offer to readers.

Sensitivity analysis of SUDO's hyperparameters

To encourage the adoption of SUDO, we conducted several experiments on the Flatiron Health ECOG PS dataset to measure SUDO's sensitivity to hyperparameters. Specifically, we varied the number of data points sampled from each probability interval (Fig. 1b, Step 3), the type of classifier used to distinguish between pseudo- and ground-truth labelled data points (Fig. 1b, Step 4), and the amount of label noise in the held-out data being evaluated on (Fig. 1b, Step 5). We found that reducing the number of sampled data points (from 200 to just 50) and using different classifiers (logistic regression and random forest) continued to produce a strong correlation between SUDO and model performance ($|r| > 0.94$) (Supplementary Fig. 4). Such variations, however, altered the directionality (net positive or negative) of the SUDO values (from one experiment to the next) in the probability intervals with a high degree of class contamination. For example, in the interval $0.20 < p < 0.25$ (Fig. 3a), $\text{SUDO}_{\text{AUC}} = 0.05$ and $\text{SUDO}_{\text{AUC}} = -0.05$ when sampling 50 data points compared to 200 data points. We argue that such an outcome does not practically affect the interpretation of SUDO, as it is the absolute value of SUDO that matters most when it comes to identifying unreliable predictions. We offer guidelines on how to deal with this scenario in a later section.

SUDO is insensitive to various hyperparameters

Here, we explore the sensitivity of SUDO to various hyperparameters. We implemented SUDO on the Flatiron Health ECOG PS dataset (without ground-truth annotations) having reduced the number of data points sampled from each probability interval ($200 \rightarrow 50$, Supplementary Fig. 4b), and having changed the classifier used to distinguish between pseudo-labelled and ground-truth labelled data points (Fig. 4a - logistic regression vs. Fig. 4c - random forest). We show that these changes have a minimal effect on the correlation of SUDO with the proportion of positive instances in each probability interval.

Supplementary Figure 4. SUDO is insensitive to various hyperparameters. (a-c) SUDO values on the Flatiron Health ECOG PS dataset with ground-truth annotations (a) having sampled 200 data points from each probability interval, (b) having sampled 50 data points from each probability interval, and (c) having used a different classifier to distinguish between pseudo-labelled and ground-truth labelled data points. (d-e) Correlation between the SUDO values and the proportion of the positive instances in each probability interval.

Addition 2 – Experiments to show that SUDO is agnostic to the metric used to evaluate classifiers

We implement SUDO on the Stanford DDI dataset while simply changing the metric used to evaluate classifiers from AUC (which we predominantly used throughout the manuscript) to precision and accuracy. We show that SUDO is unperturbed by these changes to the evaluation metric, and thus providing machine learning practitioners with greater flexibility on the type of metrics that they can use for their own applications. Supplementary Material → Supplementary Note 1 → SUDO is agnostic to the metric used to evaluate classifiers.

SUDO is agnostic to the metric used to evaluate classifiers

We claimed that SUDO works well with almost any metric used to evaluate the classifiers. Although we predominantly presented results for $SUDO_{AUC}$ in the main manuscript, we show that $SUDO_{Precision}$ and $SUDO_{ACC}$, where we used classifier precision and accuracy as the evaluation metrics, correlate equally well to the proportion of positive instances in each probability interval.

Supplementary Figure 5. SUDO is agnostic to the metric used to evaluate classifiers. Correlation between SUDO values and the proportion of positive instances in each probability interval on the Stanford DDI dataset. Results are shown for the (left column) DeepDerm and (right column) HAM10000 models. SUDO values are based on the (a-b) precision and the (c-d) accuracy of the classifiers.

Addition 3 – Improved description of the SUDO framework

To avoid confusion, we have now improved the description of the SUDO framework to be more concise and better aligned with Fig. 1. Specifically, we have outlined a succinct series of steps that one would have to follow to implement SUDO (Results → Overview of the SUDO framework, page 3) and ensured that these steps are consistent with the expanded explanation provided in the Methods section (Methods → Details of SUDO framework, page 11).

Overview of the SUDO framework

SUDO is a framework that helps identify unreliable AI predictions, select favourable AI systems, and assess algorithmic bias for data in the wild without ground-truth labels. We outline the mechanics of SUDO through a series of steps (Fig. 1b).

Step 1 Deploy probabilistic AI system on data points in the wild to produce output $p \in [0, 1]$ reflecting probability of positive class for each data point.

Step 2 Generate distribution of output probabilities and discretize them into several predefined intervals (e.g., deciles).

Step 3 Sample data points in the wild from each interval and assign them a temporary class label (pseudo label). Retrieve an equal number of data points in the training set from the opposite class.

Step 4 Train a classifier to distinguish between the newly pseudo-labelled data points and those with a ground-truth label.

Step 5 Evaluate classifier on held-out set of data with ground-truth labels (e.g., using any metric such as AUC). A performant classifier provides evidence in support of the pseudo-label. However, data points in each interval may belong to multiple classes, exhibiting *class contamination*. To detect this contamination, we repeat these steps yet cycle through the different pseudo-labels.

Pseudo-label discrepancy Calculate the discrepancy between the performance of the classifiers with different pseudo labels. The greater the discrepancy between classifiers, the less class contamination there is, and the more likely that the data points belong to one class. We refer to this discrepancy as the pseudo-label discrepancy or SUDO.

Addition 4 – Improved description of related work

We have incorporated the related work suggestions made by Reviewer 2 into both the Introduction and the Discussion sections. In short, we highlight that although previous work has leveraged both labelled and unlabelled data to estimate model performance, such approaches remain model-centric (focusing exclusively on model performance). As such, they overlook the data-centric decisions that would need to be considered upon deployment of models (e.g., identifying unreliable predictions) (**Introduction, page 1, paragraph 3**) and (**Discussion, page 8, paragraph 3**).

Previous work assumes highly-confident predictions are reliable^{5,6}, even though AI systems can generate highly-confident incorrect predictions⁷. Recognizing these limitations, others have demonstrated the value of modifying AI-based confidence scores through explicit calibration methods such as Platt scaling⁸ or through ensemble models⁹. Such calibration methods, however, can be ineffective when deployed on data in the wild that exhibit distribution shift¹⁰. Regardless, quantifying the effectiveness of calibration methods would still require ground-truth labels, a missing element of data in the wild. Although another line of research focuses on estimating the overall performance of models with unlabelled data^{11,12}, it tends to be model-centric, overlooking the data-centric decisions (e.g., identifying unreliable predictions) that would need to be made upon deployment of these models, and makes the oft fallible assumption that the held-out set of data is representative of data in the wild, and therefore erroneously extends findings in the former setting to those in the latter.

Compared with previous studies, our study offers a wider range of applications for predictions on data without ground-truth annotations. These applications include identifying unreliable predictions, selecting favourable models, and assessing algorithmic bias. Although previous work has focused on estimating model performance^{11,12,18,19} and assessing algorithmic bias²⁰ using both labelled and unlabelled data, their approaches are model-centric (focusing exclusively on aggregate model performance) and therefore overlook the myriad data-centric decisions that we consider and which would need to be made upon deployment of an AI system. The same limitation holds for other studies that attempt to account for verification bias^{21,22}, a form of distribution shift brought about by only focusing on labelled data.

Addition 5 – Practical guidelines for using SUDO

We offer readers evidence-based recommendations on how to use SUDO in the hope that this encourages them to adopt SUDO for their own applications and to keep an eye out for potential use-cases in which SUDO performs best. These recommendations are based on the sensitivity experiments, those conducted on the simulated data, and more generally on the real-world datasets of different modalities (dermatology images and clinical notes). Results → Practical guidelines for using SUDO (page 7)

Practical guidelines for using SUDO

We have made the case and presented evidence that SUDO can evaluate AI systems without ground-truth annotations. We now take stock of our findings to offer practical guidelines around SUDO. First, we demonstrated that SUDO works well across multiple data modalities (images, text, simulation). We therefore recommend using SUDO irrespective of the modality of data a model is evaluated on. Second, we showed that SUDO is agnostic to the neural network architecture of the AI system being evaluated (convolutional for images, feed-forward for text). The only requirement is that the neural network returns a probabilistic value. Third, we showed that SUDO can deal with as few as 50 data points sampled from each probability interval (on the Stanford DDI dataset). Although sampling too few data points did not change the absolute value of SUDO, and thereby reliably quantifying class contamination, it did alter its directionality (negative or positive), affecting the perceived proportion of the majority class in a set of predictions. To avoid being misled by this

behaviour, we recommend sampling at least 50% of the data points in each probability interval in order to capture a representative set of predictions. We also note that the absolute value of SUDO should take precedent for determining unreliable predictions. Only if that value is large enough (i.e., low class contamination) should its directionality be considered.

Fourth, we showed that SUDO is unperturbed by an imbalance in the number of data points from each class or by the presence of a third-and-unseen class (on the simulated dataset). If data in the wild are suspected to exhibit these features, then SUDO can still be used. Fifth, we showed that SUDO is sensitive to the quality of the labels in the held-out set of data. As such, we recommend curating a dataset with minimal label noise when using SUDO. Furthermore, we showed that SUDO produces consistent results irrespective of the classifier used to distinguish between pseudo-labelled and ground-truth data points and of the metric used to evaluate these classifiers. We therefore recommend using a lightweight classifier (to speed up computation) and the metric most suitable for the task at hand.

POINT-BY-POINT RESPONSE

Reviewer 1

Summary

The authors aim to tackle an important problem that clinical ML commonly faces, which is estimating the likely performance and failure modes of models at deployment when they are probed with real-world examples. The manuscript clearly motivates the problem and explains the experimental setting in which the proposed approach, SUDO, is evaluated. I would like to share a few thoughts on why I believe the paper requires more rigorous experimentation and methodological explanations to justify claims such as "it enables the evaluation of clinical ML without ground-truth annotations."

R1 – Comment 1

The experiments were conducted on two datasets, one of which (Multi-Domain Sentiment) is a general domain dataset that does not relate to clinical data. Further, the second dataset is constructed based on assumptions that do not guarantee the presence of "in the wild" or "distribution-shift" scenarios. The authors explicitly acknowledge this in the text, stating "we hypothesize that this could result in a shift in the data distribution."

From this perspective, the experimental evidence does not seem to be sufficient to make such claims. I would recommend using publicly available domain-shift datasets to validate the proposed solution, without being limited to NLP-only datasets. Additionally, the validity of the identified problems may not persist or be applicable for the language modality when large language models (LLMs) are used; thus, validating the approach on other data modalities could broaden the scope of the proposed algorithm.

Response to R1 – Comment 1

We have now implemented SUDO on a publicly-available dataset (Stanford diverse dermatology images dataset) with documented distribution shift. We chose this dataset in order to show that SUDO achieves its goal on a dataset (a) with known distribution shift and (b) with a modality that is different from natural language. The latter would demonstrate the applicability of SUDO to multiple data modalities (dermatology images and clinical notes).

In short, we show that SUDO continues to correlate well with model performance (**Results → SUDO correlates with model performance on Stanford diverse dermatology images dataset**), informs model selection (**Results → SUDO informs model selection on Stanford diverse dermatology images dataset**), and helps assess algorithmic bias (**Results → SUDO helps assess algorithmic bias without ground-truth annotations**).

SUDO correlates with model performance on Stanford diverse dermatology images dataset

We used SUDO to evaluate predictions made on the Stanford diverse dermatology image (DDI) data¹³ ($n : 656$) (see Description of datasets). We purposefully chose two AI models (DeepDerm¹⁴ and HAM10000¹⁵) that were performant on their respective data (AUC = 0.88 and 0.92) and whose performance degraded drastically when deployed on the DDI data (AUC = 0.56 and 0.67), suggesting the presence of distribution shift.

Confirming previous findings, we found that these models struggle to distinguish between benign (negative) and malignant (positive) lesions in images. This is evident by the lack

of separability of the AI-based probabilities corresponding to the ground-truth negative and positive classes (Fig. 2a and b). We set out to determine whether SUDO can quantify this class contamination (without ground-truth labels). By following the steps of SUDO (see Overview of the SUDO framework), we found that it correlates ($\rho = -0.84$ and -0.76 for DeepDerm and HAM10000, respectively) with the proportion of positive instances in each of the chosen probability intervals (Fig. 2c and d). Such a finding, which holds even if we use a different evaluation metric for SUDO (Supplementary Fig. 5), suggests that SUDO can act as a reliable proxy for the accuracy of predictions. Notably, this ability holds irrespective of the underlying performance of the AI model being evaluated, as evidenced by the high correlation values for the two models with different performance metrics (AUC = 0.56 and 0.67).

SUDO informs model selection with Stanford diverse dermatology images dataset

As a proxy for the accuracy of AI predictions, SUDO can identify two tiers of predictions: those which are sufficiently reliable for downstream analyses and others which are unreliable and might require further inspection by a human expert. This creates a trade-off between the completeness of AI predictions (i.e., are *all* predictions included in downstream analyses?) and the reliability of such predictions. Ideally, both of these elements are maximized for AI predictions. We leverage this intuition and introduce the reliability-completeness curve as a way of rank ordering models when ground-truth annotations are unavailable (Fig. 2e, see Producing reliability-completeness curve for details).

We found that the ordering of the performance of the models is consistent with that presented in previous studies¹¹. Specifically, HAM10000 and DeepDerm achieve an area under the reliability-completeness curve (AURCC = 0.864 and 0.621, respectively) and, with ground-truth annotations, these models achieve (AUC = 0.67 and 0.56). We note that the emphasis here is on the relative ordering of models and not on their absolute performance. These consistent findings suggest that SUDO can help inform model selection on data in the wild without annotations.

SUDO helps assess algorithmic bias without ground-truth annotations

Algorithmic bias often manifests as a discrepancy in model performance across two protected groups (e.g., male and female patients). Traditionally, this would involve comparing AI predictions to ground-truth labels. Viewing SUDO as a proxy for model performance, we hypothesized that it can help assess such bias even without ground-truth labels. We tested this hypothesis on the Stanford DDI dataset by stratifying the AI predictions according to the skin tone of the patients (Fitzpatrick scale I-II vs. V-VI) and implementing SUDO for each of these stratified groups. A difference in the resultant SUDO values would indicate a higher degree of class contamination (and therefore poorer performance) for one group over another. Having found that $SUDO_{AUC} = 0.60$ and 0.58 for

the two groups, respectively, we supported the validity of this bias by using the ground-truth labels to calculate the negative predictive value of the predictions (NPV). With $NPV = 0.83$ and 0.78 , respectively, we found that both SUDO and the traditional approach (with ground-truth labels) identified a bias in favour of patients with a Fitzpatrick scale of I-II; the main distinction is that SUDO did not require ground-truth labels for the dataset being evaluated. Moreover, these bias findings are consistent with those reported in a previous study¹³.

R1 – Comment 2

The proposed solution (SUDO) is explained in plain language as an algorithm; however, I find there is a lack of motivation and critical reasoning regarding potential corner cases where this algorithm could fail. For instance, the authors mention that deep neural nets can make class posterior predictions with very high confidence and still fail, which could render the inference probability estimates collected on "in-the-wild" samples (external) less meaningful. For example, what would happen if the initial model assigns "class-0" to all external samples as pseudo-labels in a systematic manner? I would expect the subsequent two models to differ in terms of their predictions; however, this does not necessarily mean that the initial classification results were correct. Without reliable confidence estimates and the injection of any additional information into the system, it is difficult to comprehend how such a system would recognize what it actually doesn't know. Therefore, the study requires more theoretical justification, along with additional experimental analysis, to support the claims made.

Response to R1 – Comment 2

The reviewer poses a question about a hypothetical scenario in which the initial model assigns one and only one class to all external samples. We would like to clarify that (a) this scenario is unlikely to happen with SUDO (by design, as explained next) and (b) that the reviewer's intuition for the expected outcome of such a scenario is not exactly precise.

To address these points, notice that SUDO requires that machine learning practitioners experiment with all possible pseudo-labels (from the set of allowable classes). For example, in the event of a binary classification, the external samples are assigned to either class 0 or class 1. If the majority of the external samples do indeed belong to class 0, then model A trained with such samples pseudo-labelled as class 0 will perform better than model B trained with the same samples pseudo-labelled as class 1.

The difference in the performance of model A and model B (which we refer to as the pseudo-label discrepancy) indicates the degree to which the external samples truly belong to either one of the classes.

Now if we decide to “assign all external samples to class 0”, as the reviewer suggested, then we would also have to conduct a corresponding experiment where we decide to assign all external samples to class 1. It is only by comparing the results of these two experiments are we able to gauge the level of contamination (i.e., ratio of class 0 vs. class 1 samples) in the external samples. If the external samples consist of, for example, a 50:50 split of class 0 and class 1 (as the ground-truth), then the downstream models which are trained on pseudo-labels of class 0 and pseudo-labels of class 1 are likely to perform (i) poorly and (ii) equally to one another. In other words, we would expect to see a low (close to zero) pseudo-label discrepancy. In fact, that is exactly what we demonstrate empirically when the confidence estimates of the initial model are between 0.4 and 0.6.

Having said that, we do acknowledge the importance of highlighting the limitations of SUDO and the edge cases where it might exhibit subpar behaviour. To that end, we have now conducted additional experiments on simulated data whereby we vary the characteristics of the data in the wild. Specifically, we vary the data in the wild to exhibit (a) distribution shift, (b) distribution shift with an imbalance in the number of data points per class, (c) distribution shift with the presence of a third-and-unseen class, and (d) distribution shift where the held-out set contains label noise.

- *Exploring the limits of SUDO with simulated data* – we show that SUDO continues to correlate well with model performance in the above scenarios (except for when label noise is present and the relationship between the data in the wild distributions is drastically changed), demonstrating its robust behaviour. These findings are presented in **Supplementary Figures 2 and 3** and summarized in **Results → Exploring the limits of SUDO with simulated data (page 5)**.

Exploring the limits of SUDO with simulated data

To shed light on the scenarios in which SUDO remains useful, we conducted experiments on simulated data that we can finely control (see Description of datasets). Specifically, we varied the data in the wild to encompass distribution shift (a) with the same two classes observed during training, (b) with a severe imbalance (8 : 1) in the number of data points from each class, and (c) alongside data points from a third and never-seen-before class. As SUDO is dependent on the evaluation of classifiers on held-out data (see Fig. 1, Step 5), we also experimented with injecting label noise into such data.

We found that SUDO continues to strongly correlate with model performance, even in the presence of a third class ($|\rho| > 0.87$, Supplementary Fig. 2). This is not surprising as SUDO is designed to simply quantify class contamination in each probability interval, regardless of the data points contributing to that contamination. However, we did find that SUDO requires held-out data to exhibit minimal label noise, where $\rho = 0.99 \rightarrow 0.33$ upon randomly flipping 50% of the labels in the held-out data to the opposite class. We also found that drastically changing the relationship between class-specific distributions of data points in the wild can disrupt the utility of SUDO (Supplementary Fig. 3).

- We offer readers evidence-based recommendations on how to use SUDO in the hope that this encourages them to adopt SUDO for their own applications and to keep an eye out for potential use-cases in which SUDO performs best. These recommendations are based on the sensitivity experiments, those conducted on the simulated data, and more generally on the real-world datasets of different modalities (dermatology images and clinical notes). Results → Practical guidelines for using SUDO (page 7)

Practical guidelines for using SUDO

We have made the case and presented evidence that SUDO can evaluate AI systems without ground-truth annotations. We now take stock of our findings to offer practical guidelines around SUDO. First, we demonstrated that SUDO works well across multiple data modalities (images, text, simulation). We therefore recommend using SUDO irrespective of the modality of data a model is evaluated on. Second, we showed that SUDO is agnostic to the neural network architecture of the AI system being evaluated (convolutional for images, feed-forward for text). The only requirement is that the neural network returns a probabilistic value. Third, we showed that SUDO can deal with as few as 50 data points sampled from each probability interval (on the Stanford DDI dataset). Although sampling too few data points did not change the absolute value of SUDO, and thereby reliably quantifying class contamination, it did alter its directionality (negative or positive), affecting the perceived proportion of the majority class in a set of predictions. To avoid being misled by this

behaviour, we recommend sampling at least 50% of the data points in each probability interval in order to capture a representative set of predictions. We also note that the absolute value of SUDO should take precedent for determining unreliable predictions. Only if that value is large enough (i.e., low class contamination) should its directionality be considered.

Fourth, we showed that SUDO is unperturbed by an imbalance in the number of data points from each class or by the presence of a third-and-unseen class (on the simulated dataset). If data in the wild are suspected to exhibit these features, then SUDO can still be used. Fifth, we showed that SUDO is sensitive to the quality of the labels in the held-out set of data. As such, we recommend curating a dataset with minimal label noise when using SUDO. Furthermore, we showed that SUDO produces consistent results irrespective of the classifier used to distinguish between pseudo-labelled and ground-truth data points and of the metric used to evaluate these classifiers. We therefore recommend using a lightweight classifier (to speed up computation) and the metric most suitable for the task at hand.

Reviewer 2

R2 – Comment 1

Although the authors discuss related literature in identifying unreliable AI predictions in the introduction and reverse testing and validation in the conclusion, there are important gaps in the literature review with respect to evaluating model accuracy with scarce or no labeled data. For instance, within ML there has been some work in classification accuracy and fairness assessment in the positive-unlabeled and semi-supervised settings (e.g., [1,2]). Within statistics, there is a vast literature on predictive performance evaluation in the presence of missing labeled data, with early methods taking a missing data viewpoint (e.g., [3,4]). More recent proposals target challenges relevant to the analysis of electronic health records data, including a lack of labeled data and distribution shift (e.g., [5–8]). Methods that operate without labeled data have also been previously considered (e.g., [9]). It is necessary to draw connections with these works and characterize the contribution of this paper in light of these developments.

Response to R2 – Comment 1

We thank the reviewer for bringing these articles to our attention. We have now incorporated the related work suggestions into both the Introduction and the Discussion sections. In short, we highlight that although previous work has leveraged both labelled and unlabelled data to estimate model performance, such approaches remain model-centric (focusing exclusively on model performance). As such, they overlook the data-centric decisions that would need to be considered upon deployment of models (e.g., identifying unreliable predictions) (**Introduction, page 1, paragraph 3**) and (**Discussion, page 8, paragraph 3**).

Previous work assumes highly-confident predictions are reliable^{5,6}, even though AI systems can generate highly-confident incorrect predictions⁷. Recognizing these limitations, others have demonstrated the value of modifying AI-based confidence scores through explicit calibration methods such as Platt scaling⁸ or through ensemble models⁹. Such calibration methods, however, can be ineffective when deployed on data in the wild that exhibit distribution shift¹⁰. Regardless, quantifying the effectiveness of calibration methods would still require ground-truth labels, a missing element of data in the wild. Although another line of research focuses on estimating the overall performance of models with unlabelled data^{11,12}, it tends to be model-centric, overlooking the data-centric decisions (e.g., identifying unreliable predictions) that would need to be made upon deployment of these models, and makes the oft fallible assumption that the held-out set of data is representative of data in the wild, and therefore erroneously extends findings in the former setting to those in the latter.

Compared with previous studies, our study offers a wider range of applications for predictions on data without ground-truth annotations. These applications include identifying unreliable predictions, selecting favourable models, and assessing algorithmic bias. Although previous work has focused on estimating model performance^{11, 12, 18, 19} and assessing algorithmic bias²⁰ using both labelled and unlabelled data, their approaches are model-centric (focusing exclusively on aggregate model performance) and therefore overlook the myriad data-centric decisions that we consider and which would need to be made upon deployment of an AI system. The same limitation holds for other studies that attempt to account for verification bias^{21, 22}, a form of distribution shift brought about by only focusing on labelled data.

R2 – Comment 2

The authors provide intuition for the mechanics of SUDO, but no technical arguments are provided. For example, under what conditions will SUDO provide consistent results with the “oracle method” based on evaluation with fully labeled data in the wild? These details need to be fleshed out given the importance of obtaining reliable conclusions for the three applications of interest. Simulation studies considering a broader range of settings in which the data generating mechanism is known would also assist in providing a more comprehensive evaluation of SUDO.

Response to R2 – Comment 2

We understand the importance of obtaining reliable conclusions for our applications of interest. To that end, we have taken the reviewer’s suggestion into account. Specifically, we have now conducted additional experiments on simulated data whereby we vary the characteristics of the data in the wild. Specifically, we vary the data in the wild to exhibit (a) distribution shift, (b) distribution shift with an imbalance in the number of data points per class, (c) distribution shift with the presence of a third-and-unseen class, and (d) distribution shift where the held-out set contains label noise.

- *Exploring the limits of SUDO with simulated data* – we show that SUDO continues to correlate well with model performance in the above scenarios (except for when label noise is present and the relationship between the data in the wild distributions is drastically changed), demonstrating its robust behaviour. These findings are presented in **Supplementary Figures 2 and 3** and summarized in **Results → Exploring the limits of SUDO with simulated data (page 5)**.

Exploring the limits of SUDO with simulated data

To shed light on the scenarios in which SUDO remains useful, we conducted experiments on simulated data that we can finely control (see Description of datasets). Specifically, we varied the data in the wild to encompass distribution shift (a) with the same two classes observed during training, (b) with a severe imbalance (8 : 1) in the number of data points from each class, and (c) alongside data points from a third and never-seen-before class. As SUDO is dependent on the evaluation of classifiers on held-out data (see Fig. 1, Step 5), we also experimented with injecting label noise into such data.

We found that SUDO continues to strongly correlate with model performance, even in the presence of a third class ($|\rho| > 0.87$, Supplementary Fig. 2). This is not surprising as SUDO is designed to simply quantify class contamination in each probability interval, regardless of the data points contributing to that contamination. However, we did find that SUDO requires held-out data to exhibit minimal label noise, where $\rho = 0.99 \rightarrow 0.33$ upon randomly flipping 50% of the labels in the held-out data to the opposite class. We also found that drastically changing the relationship between class-specific distributions of data points in the wild can disrupt the utility of SUDO (Supplementary Fig. 3).

- We offer readers evidence-based recommendations on how to use SUDO in the hope that this encourages them to adopt SUDO for their own applications and to keep an eye out for potential use-cases in which SUDO performs best. These recommendations are based on the sensitivity experiments, those conducted on the simulated data, and more generally on the real-world datasets of different modalities (dermatology images and clinical notes). **Results → Practical guidelines for using SUDO (page 7)**

Practical guidelines for using SUDO

We have made the case and presented evidence that SUDO can evaluate AI systems without ground-truth annotations. We now take stock of our findings to offer practical guidelines around SUDO. First, we demonstrated that SUDO works well across multiple data modalities (images, text, simulation). We therefore recommend using SUDO irrespective of the modality of data a model is evaluated on. Second, we showed that SUDO is agnostic to the neural network architecture of the AI system being evaluated (convolutional for images, feed-forward for text). The only requirement is that the neural network returns a probabilistic value. Third, we showed that SUDO can deal with as few as 50 data points sampled from each probability interval (on the Stanford DDI dataset). Although sampling too few data points did not change the absolute value of SUDO, and thereby reliably quantifying class contamination, it did alter its directionality (negative or positive), affecting the perceived proportion of the majority class in a set of predictions. To avoid being misled by this

behaviour, we recommend sampling at least 50% of the data points in each probability interval in order to capture a representative set of predictions. We also note that the absolute value of SUDO should take precedent for determining unreliable predictions. Only if that value is large enough (i.e., low class contamination) should its directionality be considered.

Fourth, we showed that SUDO is unperturbed by an imbalance in the number of data points from each class or by the presence of a third-and-unseen class (on the simulated dataset). If data in the wild are suspected to exhibit these features, then SUDO can still be used. Fifth, we showed that SUDO is sensitive to the quality of the labels in the held-out set of data. As such, we recommend curating a dataset with minimal label noise when using SUDO. Furthermore, we showed that SUDO produces consistent results irrespective of the classifier used to distinguish between pseudo-labelled and ground-truth data points and of the metric used to evaluate these classifiers. We therefore recommend using a lightweight classifier (to speed up computation) and the metric most suitable for the task at hand.

R2 – Comment 3

Methods for statistical inference are neither provided nor discussed. For example, one would need procedures for statistical testing and interval estimation to evaluate algorithmic bias and to compare the area under the reliability completeness curve for two candidate models.

Response to R2 – Comment 3

Although methods for statistical inference play an important role in making decisions, our study introduces a framework (SUDO) that can be thought of as *upstream* to such statistical inference. For example, in the context of selecting one of two candidate models, SUDO provides a way of ranking these models without ground-truth annotations according to the reliability-completeness curve. Once machine learning practitioners are equipped with these curves, they can then decide on their desired statistical technique to determine whether or not the curves are statistically different from one another.

R2 – Comment 4

I found the description of SUDO difficult to follow. I was initially going back and forth between Figure 1, the “Mechanics of SUDO” section, and the Methods section. More concretely, the description in the “Mechanics of SUDO” section and the presentation in Figure 1 are somewhat incongruent. The description of the pseudo-label discrepancy is also lacking detail. There is no explicit statement of (i) how the pseudo-label discrepancy is used for the three applications of identifying unreliable AI predictions, selecting AI systems, and assessing fairness of AI predictions and (ii) why SUDO is agnostic to the choice of metric.

Response to R2 – Comment 4

To improve clarity surrounding the implementation of SUDO, we have modified the section **Results → Overview of the SUDO framework** to be more concise and better aligned with Fig. 1. We have also better aligned its expanded explanation in the Methods section with the content in the main section of the manuscript. By including a brief series of steps in the Results, readers can quickly understand

the implementation of SUDO should they want to adopt it for their own use-case. In contrast, the expanded explanation in the Methods offers readers some more intuition behind the implementation.

Overview of the SUDO framework

SUDO is a framework that helps identify unreliable AI predictions, select favourable AI systems, and assess algorithmic bias for data in the wild without ground-truth labels. We outline the mechanics of SUDO through a series of steps (Fig. 1b).

Step 1 Deploy probabilistic AI system on data points in the wild to produce output $p \in [0, 1]$ reflecting probability of positive class for each data point.

Step 2 Generate distribution of output probabilities and discretize them into several predefined intervals (e.g., deciles).

Step 3 Sample data points in the wild from each interval and assign them a temporary class label (pseudo label). Retrieve an equal number of data points in the training set from the opposite class.

Step 4 Train a classifier to distinguish between the newly pseudo-labelled data points and those with a ground-truth label.

Step 5 Evaluate classifier on held-out set of data with ground-truth labels (e.g., using any metric such as AUC). A performant classifier provides evidence in support of the pseudo-label. However, data points in each interval may belong to multiple classes, exhibiting *class contamination*. To detect this contamination, we repeat these steps yet cycle through the different pseudo-labels.

Pseudo-label discrepancy Calculate the discrepancy between the performance of the classifiers with different pseudo labels. The greater the discrepancy between classifiers, the less class contamination there is, and the more likely that the data points belong to one class. We refer to this discrepancy as the pseudo-label discrepancy or SUDO.

To improve the description of the implementation details, we have now included a section (Methods → Implementation details of SUDO experiments) which outlines the chosen hyperparameters for each of the SUDO experiments.

Implementation details of SUDO experiments

SUDO involves selecting several hyperparameters. These can include the granularity and number of probability intervals, the number of data points to sample from each probability interval, the number of times to repeat the experiment, and the type of classifier to use. We offer guidelines on how to select these hyperparameters in the Results.

Stanford DDI dataset For the DeepDerm model (Fig 2a), we selected ten mutually-exclusive and equally-sized probability intervals in the range $0 < p < 1$, and sampled 10 data points from each probability interval. For the HAM10000 model (Fig 2b), we selected ten mutually-exclusive and equally-sized probability intervals in the range $0 < p < 0.5$, and sampled 50 data points from each probability interval. In the latter setting, we chose a smaller probability range and more granular probability intervals to account for the high concentration of data points as $p \rightarrow 0$.

To amortize the cost of training classifiers as part of the SUDO experiments, we extracted image representations offline (before the start of the experiments) and stored them for later retrieval. To capture a more representative subset of the data points and obtain a better estimate of the performance of these classifiers, we repeated these experiments 5 times for each probability interval and pseudo-label. To accelerate the experiments, we used a lightweight classifier such as a logistic regression, discovering that more complex models simply increased the training overhead without altering the findings. Unless otherwise noted, we adopted this strategy for all experiments.

Multi-Domain Sentiment dataset We also selected ten mutually-exclusive and equally-sized probability intervals in the range $0 < p < 1$, and sampled 50 and 10 data points from each probability interval when dealing with a network that

was *not* overconfident and one that was trained to be overconfident. We sampled fewer data points in the latter setting because prediction probability values were concentrated at the extreme ends of the probability range ($p \rightarrow 0$ and $p \rightarrow 1$), leaving fewer data points in the remaining probability intervals. As demonstrated in the Results, SUDO can deal with such data-scarce settings.

Simulated dataset We selected ten mutually-exclusive and equally-sized probability intervals in the range $0 < p < 1$, and sampled 50 data points from each probability interval.

Flatiron Health ECOG PS dataset On the Flatiron Health ECOG Performance Status dataset with ECOG PS labels, we sampled 200 data points from each probability interval. This value was chosen to capture a sufficient number of predictions from each probability interval without having to sample with replacement. On the Flatiron Health ECOG Performance Status dataset without ECOG PS labels (data in the wild), we sampled 500 data points from each interval in the range $(0, 0.45]$ and 100 data points from each interval in the range $(0.45, 1]$. This was due to the skewed distribution of the probability values generated in such a setting (see Fig. 3b).

As for the applications of SUDO, we now introduce them through empirical experiments on the Stanford DDI dataset at the beginning of the Results section. Namely, evidence in support of SUDO's ability to identify unreliable predictions is presented in Results → SUDO correlates with model performance on Stanford diverse dermatology images dataset (page 3), SUDO's ability to inform model selection is presented in Results → SUDO informs model selection with Stanford diverse dermatology images dataset (page 3), and SUDO's ability to assess algorithmic bias is presented in Results → SUDO helps assess algorithmic bias without ground-truth annotations (page 3). **Please refer to our response to Theme 1.** Moreover, we summarize how SUDO is used for each of these application in Methods → Applications of SUDO.

R2 – Comment 5

The proposed usage of the training data, held-out data, and data in the wild sets would benefit from more justification. The setting under consideration has similarities to that in [6]. Can the authors comment on the advantage of their approach and how it relates to this approach?

Response to R2 – Comment 5

It is standard in modern machine learning to have a training, validation, and held-out set. The training set is used to learn and update the parameters of the model. The validation set is used to identify the optimal set of hyperparameters of a model, and the held-out set is withheld for one final evaluation in order to provide an estimate of the expected performance of the model upon deployment. Ordinarily, data in the wild; those encountered upon deployment of a model in the real-world, are similar to the held-out set of data, and therefore we need not concern ourselves with a potential degradation in model performance. However, if such data in the wild exhibit a shift, then it is imperative to measure model performance to ensure it is working as expected.

We thank the reviewer for bringing reference [6] to our attention. We mention the limitations of this line of research more generally in the (Introduction, page 1, paragraph 3) and (Discussion, page 8, paragraph 3).

Previous work assumes highly-confident predictions are reliable^{5,6}, even though AI systems can generate highly-confident incorrect predictions⁷. Recognizing these limitations, others have demonstrated the value of modifying AI-based confidence scores through explicit calibration methods such as Platt scaling⁸ or through ensemble models⁹. Such calibration methods, however, can be ineffective when deployed on data in the wild that exhibit distribution shift¹⁰. Regardless, quantifying the effectiveness of calibration methods would still require ground-truth labels, a missing element of data in the wild. Although another line of research focuses on estimating the overall performance of models with unlabelled data^{11,12}, it tends to be model-centric, overlooking the data-centric decisions (e.g., identifying unreliable predictions) that would need to be made upon deployment of these models, and makes the oft fallible assumption that the held-out set of data is representative of data in the wild, and therefore erroneously extends findings in the former setting to those in the latter.

Compared with previous studies, our study offers a wider range of applications for predictions on data without ground-truth annotations. These applications include identifying unreliable predictions, selecting favourable models, and assessing algorithmic bias. Although previous work has focused on estimating model performance^{11,12,18,19} and assessing algorithmic bias²⁰ using both labelled and unlabelled data, their approaches are model-centric (focusing exclusively on aggregate model performance) and therefore overlook the myriad data-centric decisions that we consider and which would need to be made upon deployment of an AI system. The same limitation holds for other studies that attempt to account for verification bias^{21,22}, a form of distribution shift brought about by only focusing on labelled data.

As for that specific study, it focuses exclusively on estimating model performance and not on the quality of predictions at a granular level. As such, they are unable to address the application area of identifying unreliable predictions. In some ways, this is the core of our study. Their study also makes the assumption that unlabelled data are available from both the source *and* target data distributions. In contrast, and to use their terminology, we are focused on unlabelled target data alone. Notably, their approach involves calculating a density ratio based on knowledge of the source and target data distributions. This ratio runs the risk of being mis-specified and can be non-trivial to compute. In contrast, SUDO does not make any assumptions about the target distribution. Lastly, and in contrast to their proposed approach, the SUDO framework is quite modular. For example, any classifier can be used as a plug-in replacement when distinguishing between pseudo- and ground-truth labelled data points. This modularity introduces a level of flexibility that machine learning practitioners may find appealing so that they can adapt SUDO to their own use-cases.

R2 – Comment 6

SUDO relies on several hyperparameter and modeling choices, including the discretion into quantiles, the number of samples from each quantile, the choice of classifier, choice of discrepancy metric and threshold, etc. Can the authors provide more discussion and/or evaluation of the sensitivity of their method to these choices? What specific guidance can be provided to practitioners?

Response to R2 – Comment 6

We implemented SUDO on the Flatiron Health ECOG PS dataset while varying the number of data points sampled from each probability interval, the type of classifier used to distinguish between pseudo and ground-truth labelled data points, and the amount of label noise in the held-out data being evaluated on. We show that SUDO continues to correlate well with model performance even when we sample as few as 10 data points from each probability interval, suggesting that it can be applied in data-scarce settings. While we also show that SUDO is agnostic to the type of classifier used (e.g., logistic regression vs. random forest), it is surely affected by the presence of label noise in the held-out set of data. Results → Sensitivity analysis of SUDO's hyperparameters (page 5). We ultimately use these findings to inform the practical guidelines we offer to readers (Results → Practical guidelines for using SUDO, page 7).

Sensitivity analysis of SUDO's hyperparameters

To encourage the adoption of SUDO, we conducted several experiments on the Flatiron Health ECOG PS dataset to measure SUDO's sensitivity to hyperparameters. Specifically, we varied the number of data points sampled from each probability interval (Fig. 1b, Step 3), the type of classifier used to distinguish between pseudo- and ground-truth labelled data points (Fig. 1b, Step 4), and the amount of label noise in the held-out data being evaluated on (Fig. 1b, Step 5). We found that reducing the number of sampled data points (from 200 to just 50) and using different classifiers (logistic regression and random forest) continued to produce a strong correlation between SUDO and model performance ($|\rho| > 0.94$) (Supplementary Fig. 4). Such variations, however, altered the directionality (net positive or negative) of the SUDO values (from one experiment to the next) in the probability intervals with a high degree of class contamination. For example, in the interval $0.20 < p < 0.25$ (Fig. 3a), $\text{SUDO}_{\text{AUC}} = 0.05$ and $\text{SUDO}_{\text{AUC}} = -0.05$ when sampling 50 data points compared to 200 data points. We argue that such an outcome does not practically affect the interpretation of SUDO, as it is the absolute value of SUDO that matters most when it comes to identifying unreliable predictions. We offer guidelines on how to deal with this scenario in a later section.

We implement SUDO on the Stanford DDI dataset while simply changing the metric used to evaluate classifiers from AUC (which we predominantly used throughout the manuscript) to precision and accuracy. We show that SUDO is unperturbed by these changes to the evaluation metric, and thus providing machine learning practitioners with greater flexibility on the type of metrics that they can use for their own applications. Supplementary Material → Supplementary Note 1 → SUDO is agnostic to the metric used to evaluate classifiers.

SUDO is agnostic to the metric used to evaluate classifiers

We claimed that SUDO works well with almost any metric used to evaluate the classifiers. Although we predominantly presented results for SUDO_{AUC} in the main manuscript, we show that SUDO_{Precision} and SUDO_{ACC}, where we used classifier precision and accuracy as the evaluation metrics, correlate equally well to the proportion of positive instances in each probability interval.

Supplementary Figure 5. SUDO is agnostic to the metric used to evaluate classifiers. Correlation between SUDO values and the proportion of positive instances in each probability interval on the Stanford DDI dataset. Results are shown for the (left column) DeepDerm and (right column) HAM10000 models. SUDO values are based on the (a-b) precision and the (c-d) accuracy of the classifiers.

R2 – Comment 7

The Multi-Sentiment dataset is used to illustrate the utility of SUDO. While I agree with the authors that this dataset is a good testing ground, the paper is motivated by the application of AI in medicine. It would be preferable for this analysis to be supplemental and for a publicly available medical dataset to be presented in the main text. For example, the MIMIC dataset (or potentially the MIMIC phenotype annotation dataset) is one option and well-known by the ML for health community.

Response to R2 – Comment 7

To demonstrate the applicability of SUDO to datasets of different modalities with a known distribution shift, we have now conducted additional experiments on the Stanford diverse dermatology images dataset. **Please refer to our response to Theme 1.**

Furthermore, to maintain the focus of the manuscript on clinical data and AI systems, we have relegated our results on non-clinical datasets (e.g., Multi-Domain Sentiment and simulated data) to the Supplementary Material. However, we still briefly summarize these findings in **Results → SUDO can even be used with overconfident models (page 4).**

SUDO can even be used with overconfident models

AI systems are prone to producing erroneous overconfident predictions, and thereby making it difficult to rely on their confidence scores alone to identify unreliable predictions. It is in these settings where SUDO would add most value. To demonstrate this, we first trained a natural language processing (NLP) model to distinguish between negative ($n : 1000$) and positive ($n : 1000$) sentiment in product reviews with distribution shift as part of the Multi-Domain Sentiment dataset¹⁴ (see Methods for description of data). We showed that SUDO continues to correlate with model performance, pointing to the applicability of the framework across data modalities. To simulate an overconfident model, we then overtrained (i.e., extended training for an additional number of epochs) the same NLP model, as confirmed by the more extreme distribution of the prediction probability values (Supplementary Fig. 1b). Notably, we found that SUDO continues to correlate well with model performance despite the presence of overconfident predictions (Supplementary Fig. 1h). This is because SUDO leverages pseudo-labels to quantify class contamination and is not exclusively dependent on confidence scores.

R2 – Comment 8

In the Multi-Domain Sentiment analysis, why are fairness evaluations not considered? Is information on protected attributes not available? If so, can a simulated variable be created?

Response to R2 – Comment 8

That is correct; attribute information is not available for the Multi-Domain Sentiment dataset. However, with the inclusion of the new dataset (Stanford DDI), we now assess the algorithmic bias of the HAM10000 model when deployed on images with a skin tone of I-II and V-VI on the Fitzpatrick scale (**Results → SUDO helps assess algorithmic bias without ground-truth annotations, page 3**). In short, we show that both SUDO and a performance metric based on ground-truth labels are able to detect algorithmic bias in favour of the skin tone I-II.

SUDO helps assess algorithmic bias without ground-truth annotations

Algorithmic bias often manifests as a discrepancy in model performance across two protected groups (e.g., male and female patients). Traditionally, this would involve comparing AI predictions to ground-truth labels. Viewing SUDO as a proxy for model performance, we hypothesized that it can help assess such bias even without ground-truth labels. We tested this hypothesis on the Stanford DDI dataset by stratifying the AI predictions according to the skin tone of the patients (Fitzpatrick scale I-II vs. V-VI) and implementing SUDO for each of these stratified groups. A difference in the resultant SUDO values would indicate a higher degree of class contamination (and therefore poorer performance) for one group over another. Having found that $SUDO_{AUC} = 0.60$ and 0.58 for

the two groups, respectively, we supported the validity of this bias by using the ground-truth labels to calculate the negative predictive value of the predictions (NPV). With NPV = 0.83 and 0.78, respectively, we found that both SUDO and the traditional approach (with ground-truth labels) identified a bias in favour of patients with a Fitzpatrick scale of I-II; the main distinction is that SUDO did not require ground-truth labels for the dataset being evaluated. Moreover, these bias findings are consistent with those reported in a previous study¹³.

R2 – Comment 9

In the first Flatiron ECOG PS analysis, there does not appear to be any bias across groups defined by gender. Can the authors provide evaluation for a protected attribute where bias is present to illustrate SUDO performs well in this setting?

Response to R2 – Comment 9

We have now conducted such an experiment in order to demonstrate that SUDO is able to detect algorithmic bias. Please refer to *Response to R2 – Comment 8* above.

R2 – Comment 10

In the survival analysis of the Flatiron ECOG PS dataset, there is a well known relationship between mortality and ECOG PS. How would one evaluate SUDO in a setting where no such knowledge exists? This is briefly mentioned in the limitations discussion at the end of the paper, but I would suggest more detail on this point.

Response to R2 – Comment 10

To validate SUDO without ground-truth annotations, we measured its correlation with median survival time, a clinical outcome with a known relationship to ECOG PS. This approach was made possible by leveraging domain knowledge. In settings where such a relationship is unknown, we recommend identifying clinical features in the labelled data that are unique to patient cohorts. These features can include the type and dosage of medication patients receive and whether or not they were enrolled in a clinical trial. A continuous feature (e.g., medication dosage) may be preferable to a discrete one (e.g., on or off medication) in order to observe a graded response with the prediction probability intervals. If identifying one such feature is difficult and time-consuming, a data-driven alternative could involve clustering patients in the labelled data according to their clinical characteristics. Distinct clusters may encompass a set of features unique to patient cohorts. Prediction on data in the wild can then be assessed based on the degree to which they share these features. On the other hand, the more severe the distribution shift, the less likely it is that features will be shared across the labelled and unlabelled data. We now include this paragraph in Discussion (page 9, paragraph 2).

To validate SUDO without ground-truth annotations, we measured its correlation with median survival time, a clinical outcome with a known relationship to ECOG PS. This approach was made possible by leveraging domain knowledge. In settings where such a relationship is unknown, we recommend identifying clinical features in the labelled data that are unique to patient cohorts. These features can include the type and dosage of medication patients receive and whether or not they were enrolled in a clinical trial. A continuous feature (e.g., medication dosage) may be preferable to a discrete one (e.g., on or off medication) in order to observe a graded response with the prediction probability intervals. If identifying one such feature is difficult and time-consuming, a data-driven alternative could involve clustering patients in the labelled data according to their clinical characteristics. Distinct clusters may encompass a set of features unique to patient cohorts. Prediction on data in the wild can then be assessed based on the degree to which they share these features. On the other hand, the more severe the distribution shift, the less likely it is that features will be shared across the labelled and unlabelled data.

R2 – Comment 11

At a more fundamental level, any method for evaluating an AI method that does not require any gold-standard labeling of the data in the wild will inevitably rely on more modeling assumptions and in turn has the potential to lead to distorted conclusions. Can the authors add a discussion on the ethical and practical considerations related to the trade-off between more time-consuming traditional evaluation vs. the proposed approach?

Response to R2 – Comment 11

We now include the following paragraph in **Discussion (page 9, paragraph 3)**.

There are also important practical and ethical considerations when it comes to using SUDO. Without SUDO, human experts would have to painstakingly annotate all of the data points in the wild. Such an approach does not scale as datasets grow in size. Moreover, the ambiguity of certain data points can preclude their annotation by human experts. SUDO offers a way to scale the annotation process while simultaneously flagging unreliable predictions for further human inspection. However, as with any AI-based framework, over-reliance on SUDO's findings can pose risks particularly related to mislabelling data points. This can be mitigated, in some respects, by choosing a more conservative operating point on the reliability-completeness curve.

R2 – Comment 12

How much variability is there in the quality of the various ECOG PS labels presented in Table 1? What does “1” mean in this table?

Response to R2 – Comment 12

That symbol was meant to reflect that “human” annotators also applied to those entries in the table. To avoid confusion, we have now replaced those symbols with the word “human”.

R2 – Comment 13

A footnote states that SUDO “trivially extends to multi-class problems.” I suggest this be detailed in the supplement.

Response R2 – Comment 13

We comment and expand upon on how SUDO could be extended to the multi-class setting in the Discussion (page 8, paragraph 5).

SUDO’s ability to identify unreliable predictions has far-reaching implications. From a clinical standpoint, data points whose predictions are flagged as unreliable can be sent for manual review by a human expert. By extension, and from a scientific standpoint, this layer of human inspection can improve the integrity of research findings. We note that SUDO can be extended to the multi-class setting (e.g., $c > 2$ classes) by cycling through all of the pseudo-labels and retrieving data points from the mutually-exclusive classes (Fig. 1b, Step 3) to train a total of c classifiers (Fig. 1b, Step 4). The main difference to the binary setting is that SUDO would now be calculated as the maximum difference in performance across all classifiers (Fig. 1b, Step 5).

R2 – Comment 14

Fairness evaluation is only considered for a binary attribute. A short discussion of evaluation across more than 2 groups would be useful given numerous metrics have been proposed in the literature.

Response to R2 – Comment 14

We comment on how to use SUDO to assess algorithmic bias for categorical attributes in Discussion (page 8, paragraph 5). In short, SUDO can be used to assess algorithmic bias across multiple groups by simply implementing SUDO for data points from each group. Bias would still manifest as a discrepancy in the SUDO value across the groups.

We note that SUDO can also be used to assess algorithmic bias across multiple groups by simply implementing SUDO for data points from each group. Bias would still manifest as a discrepancy in the SUDO value across the groups. Overall, our study offers a first step towards a framework of inferring clinical variables which suffer from low completeness in the EHR (such as ECOG PS) in the absence of explicit documentation in their charts and ground-truth labels.

R2 – Comment 15

Is the training set in Figure 1 the same data used to train the AI system? Would this lead to bias in the three applications?

Response to R2 – Comment 15

Yes, the training set in Fig. 1 is used to both train the underlying AI system *and* train the subsequent lightweight classifiers (e.g., logistic regression) to distinguish between pseudo- and ground-truth labelled data points. The lightweight classifier, however, has never seen these training data points before and so is unlikely to be biased by them.

Having said that, there is no requirement for the retrieved data points to be from the same training set as that used for the underlying AI system. The most important part is to ensure that the held-out set of data is truly mutually-exhaustive from the data being trained on. Doing so ensures we get a reliable estimate of the classifier's performance.

R2 – Comment 16

More discussion of how to interpret and assess Figures 3g and 3h would be helpful.

Response to R2 – Comment 16

We have since removed these two subfigures from the manuscript in favour of clearer explanations.

Figure 3g (in the old manuscript) was meant to depict the SUDO values for both male and female patients per probability interval, and by extension the potential algorithmic bias of the AI system. In the new version of the manuscript, we now present results related to algorithmic bias in Results → SUDO helps assess algorithmic bias without ground-truth annotations (page 3).

SUDO helps assess algorithmic bias without ground-truth annotations

Algorithmic bias often manifests as a discrepancy in model performance across two protected groups (e.g., male and female patients). Traditionally, this would involve comparing AI predictions to ground-truth labels. Viewing SUDO as a proxy for model performance, we hypothesized that it can help assess such bias even without ground-truth labels. We tested this hypothesis on the Stanford DDI dataset by stratifying the AI predictions according to the skin tone of the patients (Fitzpatrick scale I-II vs. V-VI) and implementing SUDO for each of these stratified groups. A difference in the resultant SUDO values would indicate a higher degree of class contamination (and therefore poorer performance) for one group over another. Having found that $SUDO_{AUC} = 0.60$ and 0.58 for

the two groups, respectively, we supported the validity of this bias by using the ground-truth labels to calculate the negative predictive value of the predictions (NPV). With NPV = 0.83 and 0.78, respectively, we found that both SUDO and the traditional approach (with ground-truth labels) identified a bias in favour of patients with a Fitzpatrick scale of I-II; the main distinction is that SUDO did not require ground-truth labels for the dataset being evaluated. Moreover, these bias findings are consistent with those reported in a previous study¹³.

Figure 3h (in the old manuscript) was meant to depict the reliability-completeness curve as a means to ranking, and ultimately selecting, various AI systems. In the new version of the manuscript, we now present these results on the Stanford DDI dataset in **Results → SUDO informs model selection with Stanford diverse dermatology images dataset (page 3)**.

SUDO informs model selection with Stanford diverse dermatology images dataset

As a proxy for the accuracy of AI predictions, SUDO can identify two tiers of predictions: those which are sufficiently reliable for downstream analyses and others which are unreliable and might require further inspection by a human expert. This creates a trade-off between the completeness of AI predictions (i.e., are *all* predictions included in downstream analyses?) and the reliability of such predictions. Ideally, both of these elements are maximized for AI predictions. We leverage this intuition and introduce the reliability-completeness curve as a way of rank ordering models when ground-truth annotations are unavailable (Fig. 2e, see Producing reliability-completeness curve for details).

We found that the ordering of the performance of the models is consistent with that presented in previous studies¹¹. Specifically, HAM10000 and DeepDerm achieve an area under the reliability-completeness curve (AURCC = 0.864 and 0.621, respectively) and, with ground-truth annotations, these models achieve (AUC = 0.67 and 0.56). We note that the emphasis here is on the relative ordering of models and not on their absolute performance. These consistent findings suggest that SUDO can help inform model selection on data in the wild without annotations.

R2 – Comment 17

Small typos in the text (e.g., “in” is needed after “confidence” in the first sentence of the “Validating SUDO-guided predictions with a survival analysis” section).

Response to R2 – Comment 17

We have corrected this grammatical error.

R2 – Comment 18

If I understand correctly, it also seems the point estimates for the median survival times for the two groups from SUDO vs. the ground truth are quite different. Is this expected?

Response to R2 – Comment 18

Although we do not expect the median survival times to be perfectly similar across the labelled and unlabelled data, due to potential hidden confounding factors we cannot control for, we do believe they are similar enough to suggest that these newly-identified patient cohorts correspond to low and high ECOG PS patient cohorts. As a reminder, such a qualitative analysis is meant to complement more quantitative analyses (such as the correlation between SUDO and median survival times) and increase one's confidence in the stratified patient cohorts.

R2 – Comment 19

While I do agree that unreliable predictions can be flagged for further review, I disagree with the statement that “excluding such unreliable predictions from analyses improves the integrity of research findings.” Excluding such predictions has the potential to bias results. Ideally, analysis should be conducted after further interrogation of unreliable predictions so they may be included in analysis.

Response to R2 – Comment 19

We agree that the statement “excluding such unreliable predictions from analyses improves the integrity of research findings” can be misleading. We have therefore replaced it with the following statement “SUDO's ability to identify unreliable predictions has far-reaching implications. From a clinical standpoint, data points whose predictions are flagged as unreliable can be sent for manual review by a human expert. By extension, and from a scientific standpoint, this layer of human inspection can improve the integrity of research findings.” (**Discussion, page 8, paragraph 5**)

SUDO's ability to identify unreliable predictions has far-reaching implications. From a clinical standpoint, data points whose predictions are flagged as unreliable can be sent for manual review by a human expert. By extension, and from a scientific standpoint, this layer of human inspection can improve the integrity of research findings.

R2 – Comment 20

Will any code for SUDO be made publicly available?

Response to R2 – Comment 20

Our code is currently undergoing a patent review and is thus not publicly available.

References by R2

1. Claesen M, Davis J, De Smet F, De Moor B. Assessing binary classifiers using only positive and unlabeled data. arXiv [stat.ML]. 2015. Available: <http://arxiv.org/abs/1504.06837>
2. Ji D, Smyth P, Steyvers M. Can I trust my fairness metric? assessing fairness with unlabeled data and bayesian inference. Adv Neural Inf Process Syst. 2020;33: 18600–18612.
3. Fluss R, Reiser B, Faraggi D, Rotnitzky A. Estimation of the ROC curve under verification bias. Biom J. 2009;51: 475–490.
4. Rotnitzky A, Faraggi D, Schisterman E. Doubly Robust Estimation of the Area Under the Receiver-Operating Characteristic Curve in the Presence of Verification Bias. J Am Stat Assoc. 2006;101: 1276–1288.
5. Gronsbell JL, Cai T. Semi-supervised approaches to efficient evaluation of model prediction

performance. *Journal of the Royal Statistical Society: Series B (Statistical Methodology)*. 2018. pp. 579–594. doi:10.1111/rssb.12264

6. Wang L, Wang X, Liao KP, Cai T. Semi-supervised Transfer Learning for Evaluation of Model Classification Performance. *arXiv [stat.ME]*. 2022. Available: <http://arxiv.org/abs/2208.07927>
7. Zhou D, Liu M, Li M, Cai T. Doubly Robust Augmented Model Accuracy Transfer Inference with High Dimensional Features. *arXiv [stat.ME]*. 2022. Available: <http://arxiv.org/abs/2208.05134>
8. Gronsbell J, Liu M, Tian L, Cai T. Efficient Evaluation of Prediction Rules in Semi-Supervised Settings under Stratified Sampling. *J R Stat Soc Series B Stat Methodol*. 2022;84: 1353–1391.
9. Joseph L, Gyorkos TW, Coupal L. Bayesian estimation of disease prevalence and the parameters of diagnostic tests in the absence of a gold standard. *Am J Epidemiol*. 1995;141: 263–272

Reviewers' Comments:

Reviewer #2:

Remarks to the Author:

I thank the authors for their detailed revision and thoughtful responses to my concerns.

The authors have made the following revisions:

- (1) Additional analysis of the Stanford dermatology dataset with distributional shift.
- (2) Further simulation studies assessing the impact of different types of distributional shift on the robustness of their proposed method, SUDO.
- (3) Evaluation of the impact of the hyperparameters and choice of metric on SUDO.
- (4) Updated literature review.
- (5) More detailed description of SUDO and guidelines for its use in practice.

My final comments are below.

(1) It seems from the newly added simulations that SUDO cannot handle label noise or posterior drift. I recommend including the "Exploring the limits of SUDO on simulated data" in the main text to make these points clear to readers given their importance in practice.

(2) The lack of statistical inference procedures is a major limitation and should at least be pointed out in the discussion section. Although the authors argue one can rank competing models with the reliability-completeness curve, without uncertainty estimates it is impossible to know if these rankings are trustworthy. The same issue arises when evaluating model fairness and in identifying unreliable predictions.

(3) Related to the previous point, the authors highlight (among other points) that their work differs from related literature on evaluating model performance with labeled and unlabeled data as SUDO is less model-centric. This contradicts a point in their discussion that states that "Without SUDO, human experts would have to painstakingly annotate all of the data points in the wild." This is not true with respect to model performance given the extensive literature on the topic. Arguably, these procedures are preferable to SUDO when small labeled datasets are available.

(4) While I understand the code is under patent review, it is very concerning that the simulation and data studies (with the exception of the ECOG dataset) cannot be reproduced. The lack of code may also limit the use of the method by other researchers.

Reviewer #3:

Remarks to the Author:

I am primarily responding from the perspective of the the prior Reviewer 1's comments:

"The experiments were conducted on two datasets...the second dataset is constructed based on assumptions that do not guarantee the presence of "in the wild" or "distribution-shift" scenarios...the experimental evidence does not seem to be sufficient to make such claims. I would recommend using publicly available domain-shift datasets to validate the proposed solution, without being limited to NLP-only datasets."

* The authors partially addressed this concern by adding in the Stanford Dermatology dataset, as well as synthetic data, and thereby adding positive results on new datasets and outside the text domain. However, a significantly stronger case would have been made if the authors showed these results on in-the-wild suites/benchmarks (e.g. WILDS, a very popular 10-benchmark dataset including pathology + cell imaging), to take away any possibility of cherry-picking. Given that this work is primarily empirical and how vast/varied distribution shift can be, it would be helpful for readers to feel confidence in extrapolating to further, real-world datasets.

"I find there is a lack of motivation and critical reasoning regarding potential corner cases where

this algorithm could fail...Therefore, the study requires more theoretical justification, along with additional experimental analysis, to support the claims made."

The authors also made progress towards this point by the addition of synthetic experiments, which showed the strength and weaknesses of SUDO, including an example in which the distribution shift renders the method less meaningful. In the paper, the authors say "We also found that drastically changing the relationship between class-specific distributions of data points in the wild can disrupt the utility of SUDO". In Supplementary Figure 3, the authors state "Although this scenario may be somewhat fictitious, it helps identify when SUDO should not be depended on."

I appreciate these asides, but I believe there may be a nontrivial space of distribution shifts in which SUDO may not be reliable, in addition to the example provided by the authors. It's not clear to me how a single synthetic counterexample provides sufficient guidance to users on when the method might otherwise fail. These statements are also tucked away in the paper, and would be easy for readers to miss.

Overall, distribution shift is a vast and diverse problem (e.g. spanning subpopulation shift, domain shift, covariate shift), and some of the paper's claims are somewhat lofty ("SUDO is a reliable proxy for model performance"), given the limited number of empirical experiments and lack of accompanying theory. I do believe the method could still be empirically useful, but more experimental justification would be required.

We would like to thank the reviewers for taking the time and effort to read our manuscript and for providing us with additional valuable feedback.

POINT-BY-POINT RESPONSE

Reviewer 2

Summary

I thank the authors for their detailed revision and thoughtful responses to my concerns.

The authors have made the following revisions:

1. Additional analysis of the Stanford dermatology dataset with distributional shift.
2. Further simulation studies assessing the impact of different types of distributional shift on the robustness of their proposed method, SUDO.
3. Evaluation of the impact of the hyperparameters and choice of metric on SUDO.
4. Updated literature review.
5. More detailed description of SUDO and guidelines for its use in practice.

R2 – Comment 1

It seems from the newly added simulations that SUDO cannot handle label noise or posterior drift. I recommend including the “Exploring the limits of SUDO on simulated data” in the main text to make these points clear to readers given their importance in practice.

Response to R2 – Comment 1

Indeed, the experiments we conducted on the simulated data demonstrated, amongst other things, that SUDO is rendered less meaningful in the presence of drastic label noise and when the class-specific distributions of the data in the wild change. We had already included these results in the main text in the section **Results → Exploring the limits of SUDO with simulated data (page 5)**.

In light of the practical importance of these findings, and the extent to which they can inform the use of SUDO by practitioners, we now reiterate them in the modified version of the manuscript in the **Discussion section (page 9, paragraph 3)**.

Furthermore, despite having presented evidence of SUDO’s utility on multiple real-world datasets with distribution shift, we have not explored how SUDO would behave for the entire space of possible distribution shifts. It therefore remains an open question whether a particular type of distribution shift will render SUDO less meaningful. On some simulated data, for example, we found that SUDO is less meaningful upon introducing drastic label noise or changing the class-specific distributions of the data points in the wild. More generally, we view SUDO merely as one of the first steps in informing decision-making processes. Subsequent analyses, such as statistical significance tests, would be needed to gain further confidence in the resulting conclusions.

R2 – Comment 2

The lack of statistical inference procedures is a major limitation and should at least be pointed out in the discussion section. Although the authors argue one can rank competing models with the reliability-completeness curve, without uncertainty estimates it is impossible to know if these rankings are trustworthy. The same issue arises when evaluating model fairness and in identifying unreliable predictions.

Response to R2 – Comment 2

While we agree with the reviewer about the importance of statistical inference procedures, we view SUDO merely as one of the first steps a practitioner would take to help inform downstream decisions (whether they be identifying unreliable predictions, assessing model fairness, or ranking competing models). Subsequent analyses, such as statistical significance tests, would then be needed to confirm these findings and gain additional confidence in their validity. In the modified version of the manuscript, we have mentioned this in the **Discussion section (page 9, paragraph 3)**.

More generally, we view SUDO merely as one of the first steps in informing decision-making processes. Subsequent analyses, such as statistical significance tests, would be needed to gain further confidence in the resulting conclusions.

R2 – Comment 3

Related to the previous point, the authors highlight (among other points) that their work differs from related literature on evaluating model performance with labeled and unlabeled data as SUDO is less model-centric. This contradicts a point in their discussion that states that “Without SUDO, human experts would have to painstakingly annotate all of the data points in the wild.” This is not true with respect to model performance given the extensive literature on the topic. Arguably, these procedures are preferable to SUDO when small labeled datasets are available.

Response to R2 – Comment 3

To be more precise about our use of the phrase “less model-centric”, we mean that SUDO provides you with the optionality of making decisions about either a set of data points (e.g., whether their associated predictions are unreliable) or a set of models (e.g., whether their performance is different relative to one another). SUDO is not restricted to making decisions exclusively about model performance. We have clarified this in the **Discussion section (page 8, paragraph 3)**.

Previous work tends to be more model-centric than SUDO, focusing on estimating model performance^{11,12,20,21} and assessing algorithmic bias²² using both labelled and unlabelled data. It therefore overlooks the myriad data-centric decisions that would need to be made upon deployment of an AI system, such as identifying unreliable predictions. The same limitation holds for other studies that attempt to account for verification bias^{23,24}, a form of distribution shift brought about by only focusing on labelled data. In contrast, SUDO provides the optionality of guiding decisions at the model level (e.g., relative model performance) and at the data level (e.g., identifying unreliable predictions).

Having said that, the reviewer's comment about the utility of SUDO in various data regimes (i.e., small vs. large datasets) has encouraged us to touch upon it in the Discussion section. We view the utility of SUDO in identifying unreliable predictions as greatest when presented with too large a number of unlabelled data points in the wild for a team of annotators to manually annotate. In such a setting, SUDO can be viewed as a data triage mechanism, funnelling the most unreliable predictions to human annotators. In doing so, it stands to reduce the annotation burden placed on such annotators. From a practical standpoint, in settings with a handful of unlabelled data points in the wild, it might just be preferable to annotate those data points manually than to employ SUDO (see Discussion, page 9, paragraph 3).

It is also worth noting that SUDO may be considered excessive if the amount of data in the wild is small and can be annotated by a team of experts with reasonable effort. However, when presented with large-scale data in the wild, SUDO can yield value by acting as a data triage mechanism, funneling the most unreliable predictions for further inspection by human annotators. In doing so, it stands to reduce the annotation burden placed on such annotators.

R3 – Comment 4

While I understand the code is under patent review, it is very concerning that the simulation and data studies (with the exception of the ECOG dataset) cannot be reproduced. The lack of code may also limit the use of the method by other researchers.

Response to R3 – Comment 4

In the spirit of disseminating SUDO to a wide group of researchers, and after much internal deliberation, we have decided to open-source our codebase to the public. This will allow researchers to reproduce the experiments we have conducted on publicly-available datasets (e.g., Multi-Domain Sentiment, Stanford DDI, Camelyon17-WILDS, etc.) and to use SUDO on their own datasets. We have updated the Code Availability (page 15) statement accordingly.

Code availability

Our code is publicly available and can be accessed at <https://github.com/flatironhealth/SUDO>.

Reviewer 3 (replacement for Reviewer 1)

R3 – Comment 1

I am primarily responding from the perspective of the prior Reviewer 1's comments:

"The experiments were conducted on two datasets...the second dataset is constructed based on assumptions that do not guarantee the presence of "in the wild" or "distribution-shift" scenarios...the experimental evidence does not seem to be sufficient to make such claims. I would recommend using publicly available domain-shift datasets to validate the proposed solution, without being limited to NLP-only datasets." [Comment from previous R1]

The authors partially addressed this concern by adding in the Stanford Dermatology dataset, as well as synthetic data, and thereby adding positive results on new datasets and outside the text domain.

However, a significantly stronger case would have been made if the authors showed these results on in-the-wild suites/benchmarks (e.g. WILDS, a very popular 10-benchmark dataset including pathology + cell imaging), to take away any possibility of cherry-picking. Given that this work is primarily empirical and how vast/varied distribution shift can be, it would be helpful for readers to feel confidence in extrapolating to further, real-world datasets.

Response to R3 – Comment 1

We appreciate, and agree with, the reviewer's comment about instilling readers with additional confidence in the capabilities of SUDO. To do so, we had conducted experiments on three datasets (the Stanford DDI dataset, the Amazon Multi-Domain Sentiment dataset, and the Flatiron Health ECOG dataset). Each of these datasets exhibits some form of distribution shift, and the entire study was primarily inspired by a challenge we faced on the real-world Flatiron Health ECOG dataset.

Nonetheless, we have taken the reviewer's suggestion of conducting an additional experiment on a benchmark dataset explicitly designed to reflect distribution shift. Specifically, we implement SUDO on the **Camelyon17-WILDS benchmark dataset** (which is a part of the WILDS benchmark). We show, as with the other datasets, that SUDO correlates with model performance on the Camelyon17-WILDS histopathology dataset. Given that the distribution shift present in this dataset is driven by data from different hospitals, our finding should provide readers with the additional confidence in the utility of SUDO (see SUDO correlates with model performance on Camelyon17-WILDS histopathology dataset, page 4, paragraph 1).

SUDO correlates with model performance on Camelyon17-WILDS histopathology dataset

We provide further evidence that SUDO can identify unreliable predictions on datasets that exhibit distribution shift. Here, we trained a model on the Camelyon17-WILDS dataset to perform binary tumour classification (presence vs. absence) based on a single histopathological image, and evaluated the predictions on the corresponding test set ($n : 85,054$) (see Description of datasets). This dataset has been constructed such

that the test set contains data from a hospital unseen during training, and is thus considered *in the wild*.

We found that the trained model achieved an average accuracy ≈ 0.85 despite being presented with images from an unseen hospital. This is supported by the relative separability of the class-specific distributions of the AI-based probabilities (Fig. 3a). We then used SUDO to quantify the class contamination at various probability intervals (Fig. 3b), finding that it continues to correlate ($\rho = -0.79$) with the proportion of positive instances in each of the chosen probability intervals.

R3 – Comment 2

"I find there is a lack of motivation and critical reasoning regarding potential corner cases where this algorithm could fail...Therefore, the study requires more theoretical justification, along with additional experimental analysis, to support the claims made." [Comment from previous R1]

The authors also made progress towards this point by the addition of synthetic experiments, which showed the strength and weaknesses of SUDO, including an example in which the distribution shift renders the method less meaningful. In the paper, the authors say "We also found that drastically

changing the relationship between class-specific distributions of data points in the wild can disrupt the utility of SUDO". In Supplementary Figure 3, the authors state "Although this scenario may be somewhat fictitious, it helps identify when SUDO should not be depended on."

I appreciate these asides, but I believe there may be a nontrivial space of distribution shifts in which SUDO may not be reliable, in addition to the example provided by the authors. It's not clear to me how a single synthetic counterexample provides sufficient guidance to users on when the method might otherwise fail. These statements are also tucked away in the paper, and would be easy for readers to miss.

Overall, distribution shift is a vast and diverse problem (e.g. spanning subpopulation shift, domain shift, covariate shift), and some of the paper's claims are somewhat lofty ("SUDO is a reliable proxy for model performance"), given the limited number of empirical experiments and lack of accompanying theory. I do believe the method could still be empirically useful, but more experimental justification would be required.

Response to R3 – Comment 2

While distribution shift does come in many forms, as the reviewer aptly points out, we can only reasonably evaluate our approach on a handful of datasets that we believe reflect the scenarios of interest. As outlined above, this entire study was inspired by the challenges we encountered with the real-world Flatiron Health ECOG dataset. By conducting experiments on a multitude of datasets (Stanford DDI, Amazon Multi-Domain Sentiment) and in the latest revision, the Camelyon17-WILDS dataset, we believe we have provided the additional experimental justification for SUDO (see Response to R3 – Comment 1).

As for the lofty language, we have now minimized these instances throughout the manuscript. For example, instead of communicating that "SUDO is a reliable proxy...", we now say "SUDO can be a reliable proxy...", illustrating that SUDO works well in the settings we have presented, but that it is entirely possible that it might fail in other as-of-yet unexplored settings. Furthermore, to provide a more balanced narrative of the utility of SUDO, we have expanded the limitations sub-section of the Discussion section to account for the reviewer's comments (Discussion, page 9, paragraph 3).

ability intervals. It is also worth noting that SUDO may be considered excessive if the amount of data in the wild is small and can be annotated by a team of experts with reasonable effort. However, when presented with large-scale data in the wild, SUDO can yield value by acting as a data triage mechanism, funneling the most unreliable predictions for further inspection by human annotators. In doing so, it stands to reduce the annotation burden placed on such annotators. Furthermore, despite having presented evidence of SUDO's utility on multiple real-world datasets with distribution shift, we have not explored how SUDO would behave for the entire space of possible distribution shifts. It therefore remains an open question whether a particular type of distribution shift will render SUDO less meaningful. On some simulated data, for example, we found that SUDO is less meaningful upon introducing drastic label noise or changing the class-specific distributions of the data points in the wild. More generally, we view SUDO merely as one of the first steps in informing decision-making processes. Subsequent analyses, such as statistical significance tests, would be needed to gain further confidence in the resulting conclusions.

Reviewers' Comments:

Reviewer #2:

Remarks to the Author:

The authors have addressed all remaining concerns and have released their code as requested.